# Sustainable 3D printing by reversible salting-out effects with aqueous salt solutions

Donghwan Ji ®[1], Joseph Liu ®[1], Jiayu Zhao[1], Minghao Li ®[2], Yumi Rho ®[1,3], Hwansoo Shin ®[4], Tae Hee Han ®[4] & Jinhye Bae ®[1,2,3] ✉

Achieving a simple yet sustainable printing technique with minimal instruments and energy remains challenging. Here, a facile and sustainable 3D printing technique is developed by utilizing a reversible salting-out effect. The salting-out effect induced by aqueous salt solutions lowers the phase transition temperature of poly(N-isopropylacrylamide) (PNIPAM) solutions to below 10 °C. It enables the spontaneous and instant formation of physical crosslinks within PNIPAM chains at room temperature, thus allowing the PNIPAM solution to solidify upon contact with a salt solution. The PNIPAM solutions are extrudable through needles and can immediately solidify by salt ions, preserving printed structures, without rheological modifiers, chemical crosslinkers, and additional post-processing steps/equipment. The reversible physical crosslinking and de-crosslinking of the polymer through the salting-out effect demonstrate the recyclability of the polymeric ink. This printing approach extends to various PNIPAM-based composite solutions incorporating functional materials or other polymers, which offers great potential for developing water-soluble disposable electronic circuits, carriers for delivering small materials, and smart actuators.

3D printing of polymers enables rapid prototyping and construction of small-scale complex structures at a relatively lower cost compared to traditional subtractive manufacturing (i.e., removal of materials) or formative manufacturing (i.e., reshaping/molding of materials) methods[1–4]. Among the various additive manufacturing techniques, such as fused deposition modeling (FDM)[5,6], stereolithography (SLA)[7,8], selective laser sintering (SLS)[9,10], and direct ink writing (DIW)[11,12], the DIW method has gained extensive attention owing to its cost-effectiveness with simple procedures and fewer energies and a relatively wider range of material selections[13,14]. The DIW works by extruding inks with mechanical or pneumatical force out of a syringe needle/nozzle onto a solid substrate. These inks should have viscoelastic and shear-thinning properties for smooth extrusion and have sufficiently high viscosity in the range of a few mPa s to kPa s for shape retention after the extrusion[15,16]. To achieve such good printability and structural integrity of the extruding inks,

rheological modification and post-processing steps (e.g., chemical crosslinking under heat or light), respectively, are generally employed. However, the rheological modification often provokes a component precipitation or flocculation due to their sensitivity to the ionic strength of ink precursors as well as an increment in the complexity of the 3D printing process[16,17]. Post-processing usually requires additional equipment, energy, and catalysts along with chemical crosslinkers (e.g., crosslinkers with aldehyde, cyanoacrylate, or epoxide groups), which may adversely affect human health and environments[18]. Furthermore, the resultant crosslinked structures suffer from recycling due to the limitation in dissolving or decomposing the structures through aqueous-based systems sustainably[19,20]. Therefore, while achieving good printability and structural integrity of the printed shape, different approaches enabling 3D printing need to be developed in a simple and sustainable system/process[21].

[1]Department of NanoEngineering, University of California San Diego, La Jolla, CA 92093, USA. [2]Materials Science and Engineering Program, University of California San Diego, La Jolla, CA 92093, USA. [3]Chemical Engineering Program, University of California San Diego, La Jolla, CA 92093, USA. [4]Department of Organic and Nano Engineering and Human-Tech Convergence Program, Hanyang University, Seoul 04763, Republic of Korea. ✉e-mail: j3bae@ucsd.edu

The solubility of polymers, including proteins and colloids with hydrophilic and hydrophobic domains, is influenced by salt ions, a phenomenon known as the Hofmeister effect[22,23]. Certain salt ions can be strongly hydrated, stealing water molecules from the polymers and thus leading to a salting-out effect[24–27]. As a result, the addition of such salts to a polymeric solution causes precipitation, aggregation, or gelation of the polymeric molecules/chains through an increase in hydrogen bonds or hydrophobic interactions among the molecules/chains[28–30]. Recent studies have highlighted mechanically enhanced polymeric materials comprising gelatin[31], cellulose nanofibrils[32], or poly(vinyl alcohol)[33,34] by employing the salting-out effect to induce the formation of intermolecular physical crosslinks, such as hydrogen bonds and hydrophobic interactions.

In this study, we apply the salting-out effect to implement a sustainable 3D printing technique without requiring rheological modifiers, chemical crosslinkers, and additional post-processing steps/equipment. The salting-out effect that is induced by aqueous salt solutions demonstrates the spontaneous formation of physical crosslinks (i.e., hydrophobic interactions) between poly(N-isopropylacrylamide) (PNIPAM) chains and consequential solidification of aqueous PNIPAM solutions. The PNIPAM solutions are extrudable through syringe needles without high-extruding force due to their low viscosity in the magnitude of $10–100$ Pa·s and shear-thinning behavior. In particular, the PNIPAM solution that inherently undergoes a phase transition from the coil-to-globule state at lower critical solution temperature (LCST) is substantially influenced by salt ions. For instance, a $CaCl_2$ solution of concentrations exceeding 2 M lowers the LCST below 10 °C, in which the salt ions facilitate the dehydration and phase transition of PNIPAM. This salting-out effect consequently enables the extruded PNIPAM solution inks to immediately undergo the phase transition and solidification with physical crosslinking formation within PNIPAM chains, upon contact with a salt solution at room temperature (~22 °C). The formation of physical crosslinks and the solidification, resulting from the salting-out effect, is consistently observed even when the PNIPAM solution contains functional materials (e.g., hydrophilic food dye, hydrophilic MXene, and hydrophobic carbon nanotube (CNT)) or other polymers (e.g., polyvinyl alcohol (PVA), polyacrylamide (PAM), and alginate (Alg)). This printing technique, therefore, eliminates the need for rheological modifiers and chemical crosslinkers in the printing ink and the need for post-curing steps, specialized equipment, and high energy and cost. Moreover, the printed structures are easily dissolved in water and recyclable, representing this printing technique as a simple and sustainable system. To our knowledge, the salting-out effect has not previously been utilized for 3D printing. This unprecedented printing approach using the PNIPAM-based system further demonstrates the strong potential for fabricating several structures/devices, such as a water-soluble disposable electronic circuit, a carrier for delivering functional materials, and an actuator with multimodal shape morphing that responds to salt conditions.

## Results

### Salting-out effects on phase transition of PNIPAM solution

The PNIPAM chains are in a state of solvated by water molecules below the phase transition temperature (i.e., LCST); in contrast, at elevated temperatures above LCST, the PNIPAM chains lose dipole-dipole and hydrogen-bonding interactions with water molecules and then release bound water, thereby aggregating with the formation of hydrophobic interactions (Fig. 1a)[35–38]. These processes induce a phase transition of PNIPAM from the coil-to-globule state and the solidification of PNIPAM solutions. In practice, 1 M PNIPAM solution solidified and turned into opaque white through the phase transition when heated above LCST, typically 30–33 °C (Fig. 1b).

This phase transition and solidification can be induced by salt ions causing the salting-out effect, even at room temperatures rather than

elevated temperatures (Fig. 1c)[25–27]. For example, the 1 M PNIPAM solution in the coil state spontaneously solidified in the globule state at room temperature (22 °C) upon the addition of a $CaCl_2$ solution with a concentration of 2 M or higher (Fig. 1d). The solidified opaque white sample comprising aggregated and physically crosslinked PNIPAM chains returned to the coil state upon cooling to subzero temperatures, thereby becoming a transparent solution reversibly. Specifically, the solidified PNIPAM within 2 M $CaCl_2$ solution was observed to return to a transparent solution at subzero (Fig. 1d, left); and the solidified PNIPAM within 4 M $CaCl_2$ solution returned to a transparent solution at a substantially lower temperature ~–18 °C (Fig. 1d, right). This phenomenon implies that the LCST of PNIPAM is declined by the salt ions causing the salting-out effect[25–27], and the decrement of LCST varies depending on the salt ion concentration[28,30,39]. According to previous theoretical studies[25–30,39,40], the decrease in LCST of PNIPAM can be explained as follows (Fig. 1c). Salt anions polarize water molecules bound to PNIPAM chains, disrupting hydrogen bonds between the PNIPAM chains and water molecules. This disruption facilitates dehydration and aggregation of PNIPAM chains, thereby forming physical crosslinks (hydrophobic interactions) between the aggregated PNIPAM chains. Salt cations simultaneously fill the spaces near the dehydrated PNIPAM chains and bind to the amide oxygen atoms of PNIPAM. Namely, the salt ions induce the coil-to-globule phase transition analogous to the temperature elevation, and the ions of sufficient amounts decrease the LCST significantly and result in the solidification of PNIPAM solutions at room temperature.

This phase transition by the salting-out effect was reversible. The liquified PNIPAM within the 4 M $CaCl_2$ (Fig. 1d, right) solidified again over the temperature elevation (Fig. 1e). The liquified PNIPAM placed on the table at room temperature was gradually solidified from edge to inside. This observation indicates that the phase transition, coil-to-globule and globule-to-coil, is a reversible reaction resulting from the reversible physical crosslinking and de-crosslinking of PNIPAM, respectively. As well as the $CaCl_2$ solution, other various salt solutions, such as 2 M NaCl, $AlCl_3$, LiCl, and $ZnBr_2$ solutions, demonstrated the solidification of the PNIPAM solution (Fig. 1f). Such solidification by salt ions did not induce PNIPAM crystallization (Supplementary Fig. 1).

This solidification caused changes in the internal structures and intermolecular interactions among PNIPAM chains. Cross-sectional scanning electron microscopy (SEM) images of lyophilized samples of the PNIPAM solution exhibited highly porous structures due to the high water content, whereas the lyophilized sample of the PNIPAM solidified within the $CaCl_2$ solution showed a non-porous dense structure (Fig. 1g). Fourier-transform infrared (FTIR) analysis further confirmed that the salt ions significantly increased interactions between PNIPAM chains in the globule state (Fig. 1h). The PNIPAM solution exhibited strong peaks corresponding to O–H stretching and bending vibrations at 3277 and 1632 $cm^{-1}$, respectively, due to numerous hydrogen bonds of water molecules[41,42]. Heating (above 40 °C) resulted in the PNIPAM dehydration and thus the appearance of peaks, which were initially hidden by hydrated water molecules, corresponding to N–H stretching, C=O stretching, and N–H bending vibrations at approximately 3374, 1623, and 1553 $cm^{-1}$, respectively[36,43,44]. The salt ions contributed to more effective dehydration than temperature elevation and to more formation of hydrophobic interactions among the PNIPAM chains, identified as peak changes in each vibration. The solidified PNIPAM by the $CaCl_2$ exhibited amplified N–H stretching peak at 3374 $cm^{-1}$, broadened C=O stretching peak toward higher wavenumber (from 1632 to 1645 $cm^{-1}$), and shifted N–H bending peak toward lower wavenumber (from 1553 to 1549 $cm^{-1}$) than those of solidified PNIPAM by heating. Therefore, the salting-out effect definitely triggers the phase transition of PNIPAM solutions, resulting in the spontaneous solidification of the PNIPAM solution with increases in intermolecular hydrophobic interactions in the globule state.

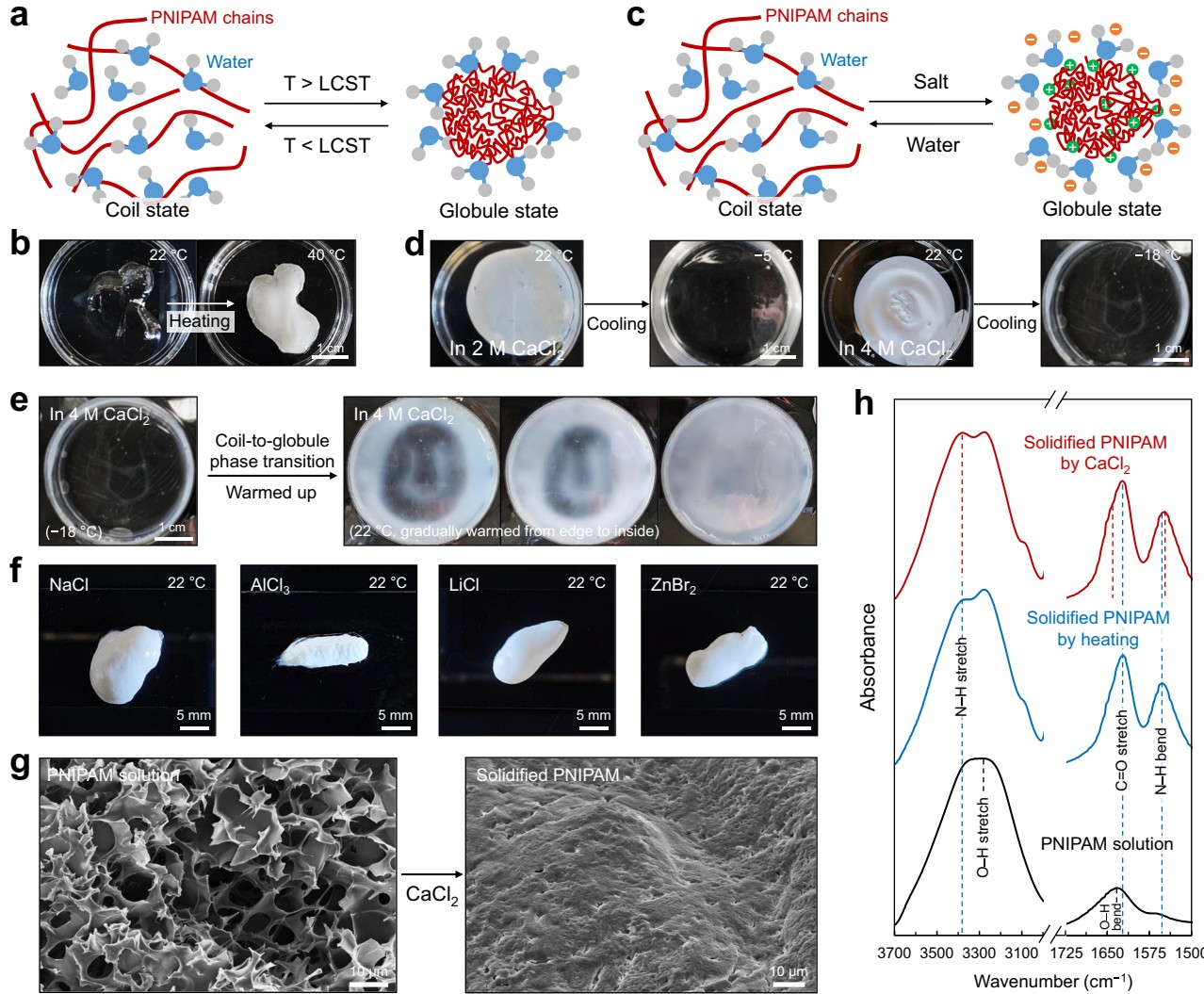

**Fig. 1 | Salting-out effect on phase transition of PNIPAM solution. a** Phase transition caused by temperature changes. **b** PNIPAM solution in the coil state below LCST and solidified PNIPAM in the globule state above LCST. **c** Phase transition caused by the salting-out effect. **d** Solidified PNIPAM within 2 M CaCl₂ solution or 4 M CaCl₂ solution, respectively, and their reversible phase transition to the coil state at subzero temperatures. **e** Re-solidification of the reversibly liquified PNIPAM within 4 M CaCl₂ solution. Before warming up the liquified solution in the coil state, it was agitated to fully disperse the PNIPAM into the CaCl₂ solution. **f** Solidification of PNIPAM solution by various salt solutions, such as 2 M NaCl, 2 M AlCl₃, 2 M LiCl, and 2 M ZnBr₂. **g** Cross-sectional SEM images of PNIPAM before/ after the phase transition caused by 2 M CaCl₂ solution. **h** FTIR patterns of PNIPAM solution and two different PNIPAM solidified by heating or 2 M CaCl₂, respectively.

## Shifts in phase transition temperature by salt ions

To clarify the effects of the presence of salts and PNIPAM concentrations on the phase transition temperature (i.e., LCST of PNIPAM solution), we examined the storage modulus change of PNIPAM over temperature elevation (Fig. 2) because gelated or solidified materials can store substantial energy and exhibit high storage modulus, unlike liquid materials. To conduct this measurement, 1 M PNIPAM that solidified within a NaCl/CaCl₂/AlCl₃ solution with different concentrations (1–4 M) was placed on an ~−20 °C rheometer stage and equilibrated until it transited to the coil state; it was then warmed at a 2 °C min⁻¹ heating rate. For the salt-free condition, the PNIPAM solution was placed on the cold stage of 10 °C and warmed up at the same heating rate.

While the PNIPAM solution without salts exhibited the phase transition near 30–31 °C, which corresponds to the onset of the storage modulus increase, the LCST of PNIPAM solution within a salt solution significantly decreased in proportion to the concentration of CaCl₂ in the range of 1–4 M (Fig. 2a). The slope of the phase separation region, especially near the onset, became steeper with a higher

concentration of CaCl₂. The greater gradient in salt concentration led to the faster penetration of ions into the sample, thus accelerating the PNIPAM dehydration[40]. Such phase transition phenomenon was consistently observed in PNIPAM with different concentrations (Fig. 2b). The PNIPAM in the concentration range from 0.3–1.3 M, which solidified within 3 M CaCl₂ solution, exhibited a similar phase transition temperature. In addition, the phase transition was reversible upon heating and cooling cycles (Fig. 2c). During these cycles, the onset temperature of the increase in storage modulus over temperature elevation and the onset temperature of the decrease in storage modulus over temperature reduction were identical. Further, the trivalent AlCl₃ and monovalent NaCl salt ions resulted in a similar phase transition trend to divalent CaCl₂ salt ions (Supplementary Fig. 2). Representatively, the PNIPAM that solidified within a 3 M NaCl, CaCl₂, or AlCl₃ solution drew almost the same storage modulus–temperature graph (Fig. 2d). The proportional decrease in LCST along the increase in NaCl, CaCl₂, or AlCl₃ concentration reveals that the phase transition was significantly influenced by the salt concentration, rather than the mono-, di-, or trivalent cation types (Fig. 2e).

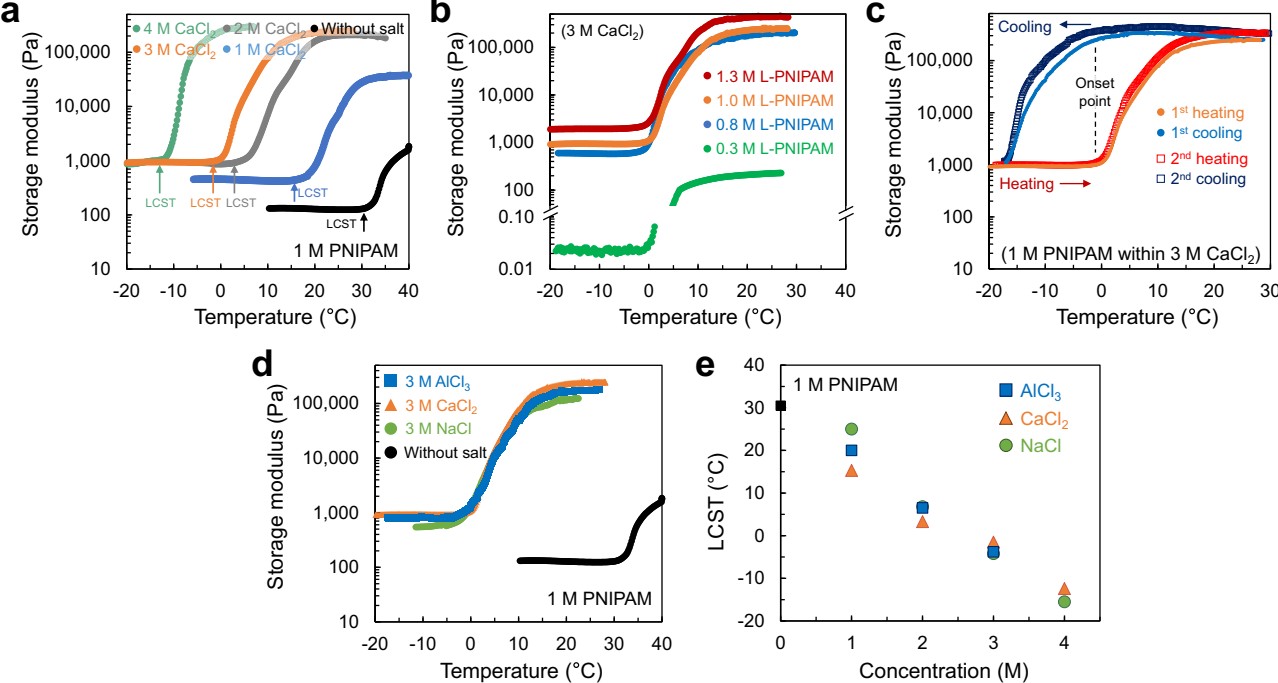

Fig. 2 | **Phase transition temperatures differed by salt ion conditions. a** Storage moduli change of 1 M PNIPAM under different CaCl$_2$ concentrations over the temperature increase. **b** Storage moduli change of PNIPAM with a different concentration at the 3 M CaCl$_2$ condition over the temperature increase. **c** Repetitive heating and cooling of 1 M PNIPAM at the 3 M CaCl$_2$ condition. **d** Storage moduli change of 1 M PNIPAM within a salt solution of different salt types, 3 M NaCl, CaCl$_2$, or AlCl$_3$. **e** Overall phase transition temperature, LCST, of 1 M PNIPAM at different salt ions and different ion concentrations.

## Spontaneous and instant solidification of PNIPAM-based solutions by reversible physical crosslinking

Based on the understanding of the salting-out effect on the PNIPAM solution and consequential solidification, we examined whether the PNIPAM-based solutions can be readily extruded through a syringe needle and spontaneously and immediately solidified upon contact with a salt solution. We also investigated whether the physically crosslinked PNIPAM can be reversibly de-crosslinked when exposed to water; water can remove salt ions between PNIPAM chains, resulting in the globule-to-coil transition (i.e., an increase in LCST beyond room temperature). For this purpose, we prepared not only a single-component pure PNIPAM solution but also various kinds of PNIPAM-based composite solutions comprising functional materials (either hydrophilic or hydrophobic additives) or other polymer materials: PNIPAM/Dye, PNIPAM/MXene, PNIPAM/CNT, PNIPAM/MXene+CNT, PNIPAM/PVA, PNIPAM/PAM, and PNIPAM/Alg.

The PNIPAM solution was extruded through a syringe needle (e.g., 20 gauge needle) and spontaneously solidified in 3 M CaCl$_2$ solution, and the extruded PNIPAM preserved its shape as extruded (Fig. 3a). The color gradient turning into opaque white indicates that the extruded PNIPAM was in the process of solidification upon contact with salt ions. To validate this spontaneous and instant solidification, we examined the storage (G') and loss (G'') moduli changes of the 1 M PNIPAM solution upon contact with different concentrations of CaCl$_2$ solution (Fig. 3b). Before the solidification, the PNIPAM solution exhibited G'' slightly larger than G'. Upon the addition of CaCl$_2$ solution to the PNIPAM solution at 120 s, G' significantly increased, surpassing G'', resulting from the phase transition and solidification of PNIPAM. The rate of changes in G' was faster with a higher concentration of CaCl$_2$ solution. Note that the actual time required to complete the solidification of the extruded solution was considerably shorter than the time duration of moduli increase shown in the rheometer test (from the addition of CaCl$_2$ solution at 120 s to 300 s in Fig. 3b). This time delay was caused by the slower diffusion of CaCl$_2$ through a

narrow gap (1000 μm truncation gap) between the rheometer stage and plate (the detailed measurement method is described in Experimental Section). The exposed area to salt ions per volume of PNIPAM solution in this measurement was constrained to be smaller than in the case of direct solution extrusion in a salt solution. The solidified sample fully dissolved in water (i.e., without salt) at room temperature (~22 °C) within a few hours, resulting from the LCST shift beyond the room temperature, which indicates the de-crosslinking of PNIPAM chains along the globule-to-coil phase transition (Fig. 3c).

Similarly, PNIPAM-based composite solutions, including PNIPAM/Dye (Fig. 3d), PNIPAM/MXene (Fig. 3e), PNIPAM/CNT (Fig. 3f), PNIPAM/PVA (Fig. 3g), PNIPAM/PAM (Fig. 3h), and PNIPAM/Alg (Fig. 3i) were readily extruded from the needle and solidified in 3 M CaCl$_2$. These extruded and solidified PNIPAM composite samples were completely dissolved in water within a few hours, except for PNIPAM/Alg (Supplementary Fig. 3). In the case of the PNIPAM/Alg solution, the solidified PNIPAM/Alg comprises interpenetrating polymer networks because Alg chains participate in crosslinking by Ca$^{2+}$ ions (Supplementary Fig. 4a)[45,46]. This solidified PNIPAM/Alg did not completely dissolve in water (Supplementary Fig. 4b), and FTIR analysis underpinned this observation (Supplementary Fig. 4c). The strong peaks corresponding to the PNIPAM chain (N–H stretch, C=O stretch, and N–H bend) and the Alg chain (O–H stretch and COO$^-$ stretch) were all identified in the solidified PNIPAM/Alg that was thoroughly rinsed in water for a few days. The instant solidification of various PNIPAM-based composite solutions was further implemented by different salt solutions, 3 M NaCl and AlCl$_3$ solutions, respectively (Supplementary Fig. 5). These results verify that adding additives, fillers, or polymers to the PNIPAM solution generally does not hinder the phase transition of PNIPAM, physical crosslinking (solidification) in salt solutions, and de-crosslinking (dissolution) in water. Exceptionally, as for a polymer (e.g., Alg) forming another type of crosslinking in addition to the physical crosslinking of PNIPAM, the interpenetrating networks can be formed and be less or not dissolved; this partial or nearly

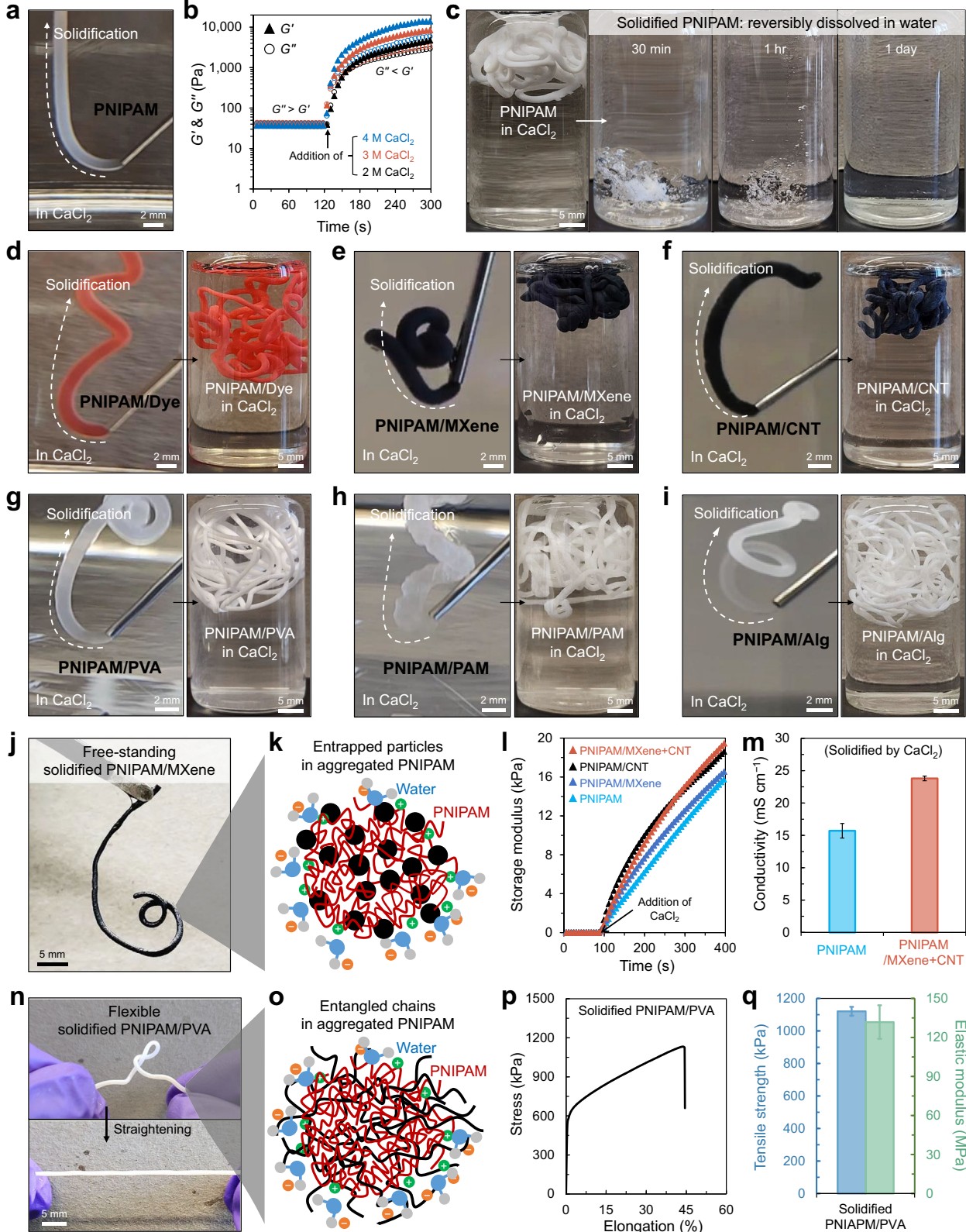

non-dissoluble PNIPAM-based system in water could be adopted for specific applications.

The solidified PNIPAM structures were free-standable overall and could have enhanced mechanical properties and/or conductivity depending on the additive component. For example, the solidified PNIPAM/MXene (Fig. 3j) demonstrated that small particles were instantly entrapped among the aggregated PNIPAM chains in the globule state before diffusing out to the surroundings (Fig. 3k). This instant aggregation was shown as an immediate increase in storage modulus upon contact of the solution with salt ions, and the composite aggregates with stiff particles resulted in higher storage modulus (Fig. 3l). Further, the solidified structure containing MXene and CNT particles demonstrated a higher conductivity than that of the solidified structure of pure PNIPAM (Fig. 3m and Supplementary Fig. 6) due to

**Fig. 3 | Instant solidification of PNIPAM-based solutions by reversible physical crosslinking. a** Extrusion and solidification of pure PNIPAM solution in 3 M CaCl₂ solution. **b** Storage (G′) and loss moduli (G″) changes of PNIPAM solution by adding 2–4 M CaCl₂ solutions. **c** Dissolution of solidified PNIPAM in water. **d–i** Extrusion and solidification of PNIPAM-based composite solutions, PNIPAM/Dye, PNIPAM/MXene, PNIPAM/CNT, PNIPAM/PVA, PNIPAM/PAM, and PNIPAM/Alg in the CaCl₂ solution, respectively. **j** Solidified PNIPAM/MXene composite displaying free-standing behavior. **k** Schematic depicting the rapid confinement of particles among the aggregated PNIPAM chains. **l** Storage modulus increase of PNIPAM and PNIPAM composite solutions upon the solidification by 3 M CaCl₂. **m** Conductivity of pure PNIPAM and PNIPAM/MXene+CNT composite solidified by 3 M CaCl₂ (n = 4). Error bars correspond to standard deviations. **n** Solidified PNIPAM/PVA composite showing deformable and stretchable behaviors. **o** Schematic illustrating the entangled polymer chains. **p** Tensile stress–strain curve and **q** the corresponding tensile strength and elastic modulus of the solidified PNIPAM/PVA in a filament form (n = 5). Error bars correspond to standard deviations.

the conductive nature of CNT and MXene particles[47–51]. In the case of the polymer composite, for instance, the solidified PNIPAM/PVA was deformable and stretchable (Fig. 3n), which contrasted with the rigid and non-stretchable structure consisting of the PNIPAM single component. This behavior was likely attributed to the entanglement of PNIPAM and PVA chains and the intrinsically good mechanical properties of PVA (Fig. 3o)[52–54]. Indeed, the solidified PNIPAM/PVA in a filament form exhibited good mechanical properties with tensile strength and elastic modulus in the magnitude of 1 MPa and 100 MPa, respectively (Fig. 3p, q).

## Rheological characteristics of PNIPAM-based solutions

We examined the rheological properties of PNIPAM solutions with a 0.5–1.6 M concentration to verify the good extrudability of the PNIPAM solutions (Fig. 4a). As a result, the PNIPAM solutions with a concentration of 1.0 M and above exhibited the shear-thinning property that has lower viscosity at high shear rates (i.e., during extruding) and higher viscosity at low shear rates (i.e., after extruding), which indicates that the PNIPAM solution with a certain concentration possesses the proper rheological property for ensuring good printability. To confirm the good extrudability of various PNIPAM composite inks, we next prepared 1.0 M PNIPAM-based composite solutions with Dye, MXene, CNT, PVA, PAM, and Alg, respectively, and these solutions exhibited the shear-thinning behavior. The addition of Dye did not considerably affect the solution viscosity, and the MXene or CNT, characterized by their strong and rigid nature, led to an increase in solution viscosity (Fig. 4b)[55,56]. The PNIPAM/polymer composite solutions also exhibited an increased viscosity (Fig. 4c). Nonetheless, all composite solutions possess the shear-thinning property regardless of the presence of additives, fillers, or polymers. The frequency sweep confirmed that PNIPAM solutions were a viscoelastic fluid overall, rather than a stable solid (Fig. 4d)[15,57]. The composite solutions also exhibited the feature of viscoelastic fluid with slightly increased moduli than those of the pure PNIPAM solution (Fig. 4d–f). This result suggests that the PNIPAM viscoelastic solutions (i.e., 1 M and above) with appropriate shear moduli and viscosity and shear-thinning behavior can be readily printed without requiring a high-extruding force. Although these solutions were in a state of liquid with G″ > G′ unlike typical solid-state printing inks with G′ > G″ (refs. 15,57), the spontaneous and instant solidification of PNIPAM-based solutions by the salting-out effect (Fig. 3) allowed us to implement 3D printing without additional rheological modifiers, chemical crosslinkers, post-processing steps, and specialized equipment.

## Sustainable 3D printing utilizing reversible salting-out effects and its potential applications

The PNIPAM solutions were applied as 3D printing inks, without rheological modifiers, chemical crosslinkers, or additional post-processing steps/equipment that are typically required for 3D printing (Fig. 5). The PNIPAM-based solution inks were printed on a salt solution-wetted glass substrate and instantly solidified (Fig. 5a). For example, the resulting printed structure of PNIPAM/CNT remained intact without the need for any post-processing or curing steps (Fig. 5a and Supplementary Fig. 7). Although we managed to print a honeycomb-patterned structure with ~10 layers using a ~200-μm-diameter nozzle (Fig. 5a), weak layer-layer adhesion between the globule

state-aggregated PNIPAM in each layer and intrinsic mechanical limitation of PNIPAM produced mechanically weak solid structures. This could be addressed by using polymer composite solution ink (Supplementary Fig. 8), such as PNIPAM/PVA forming the entanglement in polymer chains along with intrinsically good mechanical properties of PVA (Fig. 3n–q). The PNIPAM/PVA solution ink formed firm layer-layer adhesion between the solidified bottom layer containing salt ions and the freshly-printed solution top layer (Supplementary Fig. 8a–d). As a result, the multi-layered structure was free-standable and stretchable (Supplementary Fig. 8e, f). Because the PNIPAM-based solution inks exhibited the feature of low viscosity (10–100 Pa s at a shear rate of 0.1 s⁻¹), the state of almost no shear force, Fig. 4a–c) and G′ < G″ (Fig. 4d–f and Supplementary Fig. 8g) unlike conventional printing inks whose viscosity and G′ are in a few kPa–MPa s and larger than G″, respectively[2,15,16], the PNIPAM-based solution inks in extrusion-based 3D printing can be likely suitable for fabricating relatively low-height structures rather than a few cm-scale-height objects[58–60].

However, the prompt solidification by the salting-out effect allowed for an easy combination with the embedded 3D printing technique that has been known to enable freeform printing omni-directionally and the printing of complex 3D structures[61,62]. The extruded solution inks were effectively immobilized and solidified in the middle of the support bath (Fig. 5b, Supplementary Fig. 9, and Supplementary Movie 1). A physical gel serving as the support bath was prepared using an aqueous blend of Pluronic F-127 and CaCl₂ (Supplementary Fig. 10). For better visualization, the PNIPAM/CNT solution ink was extruded and immediately solidified upon contact with salt ions dissolved in the support bath; thereby, the printed structure was preserved in the middle of the bath. The PNIPAM/CNT structure that was solidified vertically in a spiral shape could be pulled out from the bath while maintaining its structural stability without failure (Supplementary Fig. 11a). The PNIPAM/PVA solution ink was also successfully printed and solidified in the middle of the bath at vertical and/or horizontal nozzle movements (Fig. 5c) and could be pulled out from the bath without fracture (Supplementary Fig. 11b). Air bubbles in the supporting bath surrounding printed structures were formed probably due to the rapid movement of the printing nozzle. These would be minimized through further control of the printing parameter and the rheology of the supporting bath.

We next investigated the recyclability of the PNIPAM solution ink (Fig. 5d). The PNIPAM ink was first printed in the shape of a recycle sign onto the flat substrate, and the printed structure was then completely dissolved by simply immersing it in water. Subsequently, the water was evaporated at a 70 °C oven, and thus, a dried PNIPAM was obtained. Finally, this dried material was re-dissolved in water to prepare the recycled ink and used for printing the recycle sign again. PNIPAM recycling was also possible by simply rinsing and dissolution in water without the drying process (Supplementary Fig. 12). The solidified PNIPAM fully dissolved in water at low temperatures (below the LCST of PNIPAM solidified by salt ions) and then was extruded into a salt solution for solidification. This solidification–dissolution cycle was repeatable. In addition to the fact that PNIPAM is widely recognized as a non-cytotoxic, non-genotoxic, and biocompatible material[63,64], this recycling of the PNIPAM solution, which consists of straightforward dissolution, collection, and re-printing procedures, demonstrates the

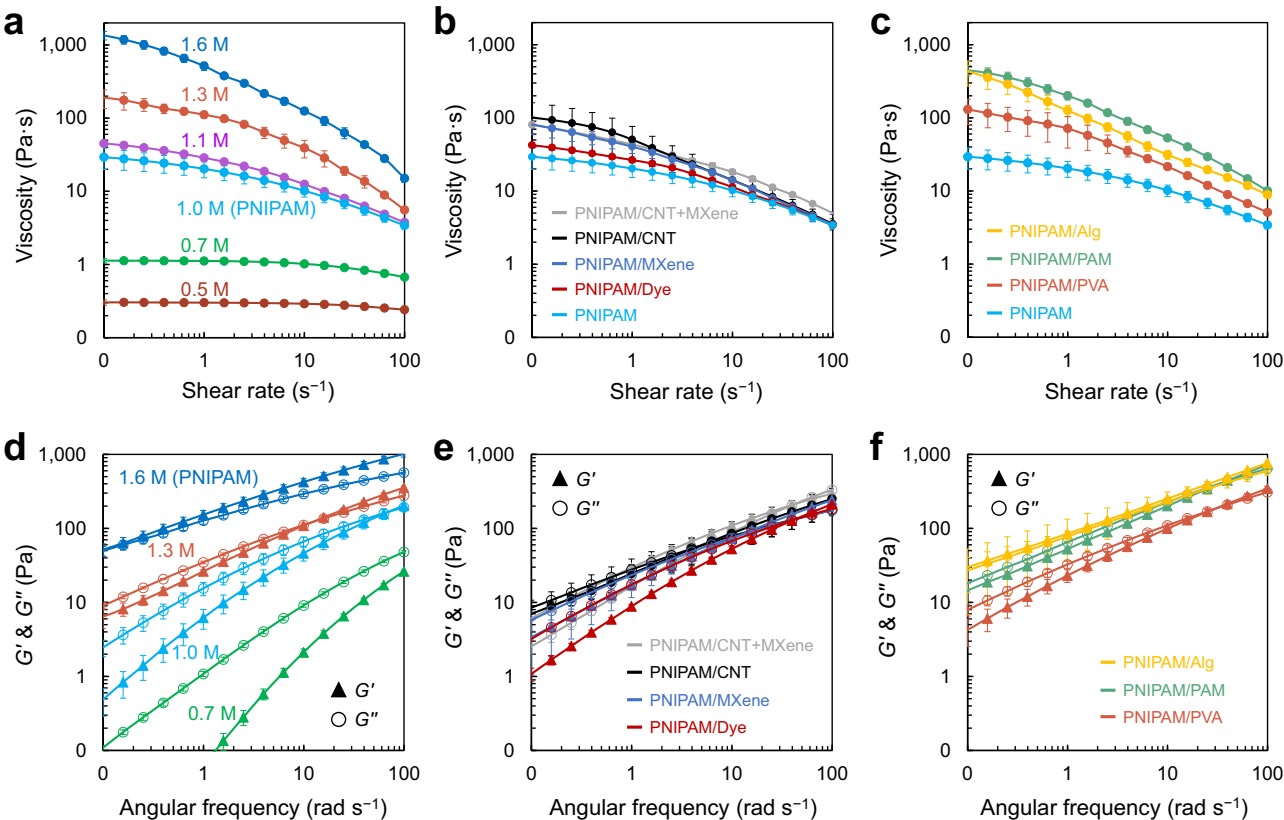

**Fig. 4 | Rheological characteristics of PNIPAM-based solutions.** Viscosity as a function of shear rate for **a** pure PNIPAM solutions with different concentrations and **b**, **c** 1.0 M PNIPAM-based composite solutions containing functional materials or other polymers, respectively. Storage ($G'$) and loss moduli ($G''$) as a function of angular frequency for **d** pure PNIPAM solutions with different concentrations and **e**, **f** 1.0 M PNIPAM-based composite solutions, respectively. Error bars correspond to standard deviations at $n \geq 3$.

potential for implementing an environmentally sustainable 3D printing.

The PNIPAM/CNT solution was applicable for printing water-soluble disposable conductive structures (Fig. 5e and Supplementary Fig. 13). The printed and dried pure PNIPAM structure failed to enable light-bulb functioning due to the lack of electrical conduction in pure PNIPAM (Supplementary Fig. 13b). In contrast, the electrical conduction through the CNTs in the PNIPAM/CNT structure, which was printed and dried, enabled a light bulb to function. Such results suggest that the PNIPAM/CNT solution ink can be used for fabricating printable electrical circuits. The brightness of this bulb gradually increased with increasing voltage from 0 to 30 V, reaching its maximum at 25–30 V (Supplementary Fig. 13c). In this demonstration, we incorporated 10% CNTs in the printed structure, and the resulting printed structure was compatible with a 25–30 V. Further studies (e.g., with different conductive particle contents) could possibly reduce the compatible voltage of the electric circuit/electrode-array for advanced electrical devices connecting with low-voltage batteries[58,59,65,66]. Meanwhile, because this printed structure comprising physically crosslinked PNIPAM was water-soluble, it demonstrates promise for developing disposable yet environmentally friendly electrically conductive structures (Fig. 5e). Moreover, the water-soluble printed structure can be utilized as a carrier delivering small particles/molecules. For example, a red dye-loaded PNIPAM structure was transferred to another substance through a simple environmental change, in this case, exposure to water (Fig. 5f). Subsequently, the dye was released upon dissolving the PNIPAM.

Furthermore, the PNIPAM/Alg solution was applicable for realizing multi-stage actuators that exhibit multiple folding angles when subjected to a solution of different salt types and concentrations.

A soft actuator, which comprised a PNIPAM/nanoclay bottom matrix and a PNIPAM/Alg hinge (Supplementary Fig. 14), demonstrated a significant fold of nearly 90° when immersed in a 3 M $CaCl_2$ solution, transitioning from its unfolded as-prepared state (180°) (Fig. 4g). This self-folding was mainly attributed to the strain mismatch between the bottom matrix and the hinge due to the significant volume contraction of the PNIPAM/Alg hinge upon the solidification by salt ions[12,67]. Subsequently, the actuator was slightly unfolded from ~90° to 115° in water, resulting from the de-crosslinking of physically crosslinked PNIPAM chains, followed by being almost unfolded in a 1 M NaCl solution caused by the de-crosslinking of Alg networks triggered by $Na^+$ ions[45,68]. This multi-stage actuator was also unfolded through a single stage in a 1 M NaCl solution without a step of soaking in water (Supplementary Fig. 15) because the 1 M NaCl solution simultaneously de-crosslinked both the $Ca^{2+}$-crosslinked Alg (Supplementary Fig. 16) and the aggregated PNIPAM chains with physical crosslinks due to the increase in the LCST of PNIPAM (~25 °C, Fig. 2e). This controllable multi- or single-stage actuation mechanism by water–salt, which has not been reported previously to our best knowledge, demonstrates another unique applicability of the reversible salting-out effects.

## Discussion

This study demonstrated that immediate aggregation of PNIPAM chains upon contact with a salt solution allowed extrusion-based 3D printing. From the perspective of interchain bonding formation, previous studies also showcased the solidification of extruded inks due to the formation of supramolecular interactions[69,70], polyelectrolyte complexes[71,72], or ionic crosslinks[73], upon contact with a certain medium. Such interchain interactions were formed in the extruded ink or between the ink and medium[74,75]. Note that we here excluded

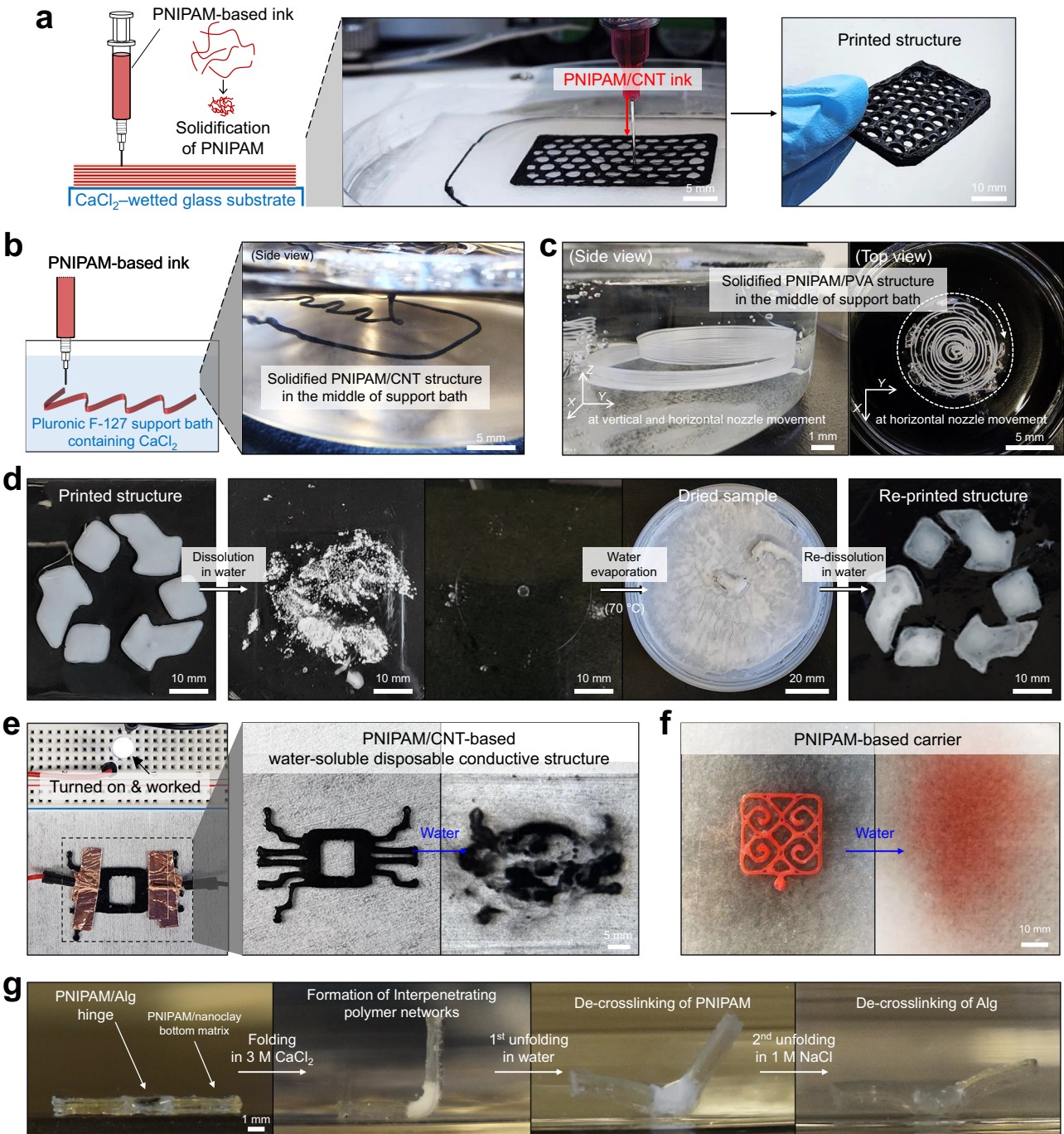

**Fig. 5 | Sustainable 3D printing utilizing reversible salting-out effects and its potential applications. a** Printing of PNIPAM-based ink solution onto a substrate wetted by a salt solution for spontaneous and rapid solidification. The right printed structure has approximately ten layers along the $Z$ axis. **b** Embedded printing of PNIPAM-based solution ink in a support bath comprising Pluronic F-127 and CaCl₂. **c** Side and top view photographs displaying two different solidified PNIPAM/PVA structures with different sizes printed in the middle of the bath. **d** The entire recycling process of the printed PNIPAM structure: dissolution in water, water evaporation, and re-dissolution in water. **e** PNIPAM/CNT-based electrically conductive and water-soluble disposable printed structure. **f** Dye-loaded PNIPAM carrier transferring the red dye to the bottom substrate while dissolving in water. **g** Self-folding and unfolding multi-stage actuator responding to simple environmental changes in salt concentrations.

discussion on typical methods requiring post-processing steps for solidification, such as chemical crosslinking under heat or light for solidification. In particular, the formation of polyelectrolyte complexes between polyelectrolyte chains with different anionic and cationic groups upon contact with water or alcohol-water mixture can be analogous to the mechanism responsible for the formation of intermolecular hydrophobic interactions in the globule state of PNIPAM by salt ions[69,70]. In contrast to the previous studies, the distinctiveness of the salting-out-based solidification strategy lies in its applicability to various PNIPAM-based composite solutions with functional particles (i.e., hydrophilic and/or hydrophobic additives) or polymeric materials. The functional particles and polymers were well mixed with the PNIPAM solution, and the PNIPAM-based composite solutions exhibited proper rheological properties and good printability. Moreover, the salting-out-based solidification was applicable to implement embedded 3D printing[74,75]. The printing of PNIPAM-based

solution ink inside a support bath with dissolved salt ions demonstrated the fabrication of the stable 3D solid structure.

In summary, we have utilized aqueous salt solutions to lower the LCST of PNIPAM, thus forming physical crosslinks among PNIPAM chains spontaneously (i.e., the salting-out effect on PNIPAM). This led to immediate solidification of the PNIPAM solution while printing at ambient temperatures (e.g., 20–25 °C). The PNIPAM solution and various PNIPAM-based composite solutions containing functional additives or polymeric materials were readily printable through syringe needles and solidified rapidly upon contact with salt ions, without requiring rheological modifiers, chemical crosslinkers, or additional post-processing steps/equipment. Furthermore, the reversible physical crosslinking and de-crosslinking of polymers via the salting-out effect facilitated the recycling of the PNIPAM solution ink, demonstrating the potential for implementing sustainable 3D printing. Such an unprecedented printing approach using the PNIPAM-based system demonstrated strong potential for a wide range of applications, for instance, in the development of a water-soluble disposable recyclable electric circuit, a smart carrier for material delivering, and a multi-stage soft actuator capable of responding to environmental changes in salt concentrations on demand without requiring chemical modifications or compatibility constraints. In terms of electrically conductive structures, we could further conduct studies on the effect of the content of conductive particles (e.g., CNT and MXene particles) and on the way of obtaining homogeneous conductive PNIPAM solutions (Supplementary Fig. 17) to increase electrical conductivity for advanced electrical devices (e.g., bioelectronics). We anticipate that this technique, employing reversible physical crosslinking and de-crosslinking of polymers through the salting-out effect, will contribute to the expansion of environmentally friendly polymer manufacturing technologies, encompassing 3D printing methodologies, recyclable polymeric devices, and smart actuators/sensors.

## Methods
### Materials
N-isopropylacrylamide (NIPAM, >98.0%, stabilized with 4-methoxyphenol) and sodium chloride (NaCl, assay >100%) were purchased from Tokyo Chemical Industry (TCI) and MP Biomedicals, respectively. N,N,N′,N′-tetramethylethylenediamine (TMEDA), ammonium persulfate (APS), calcium chloride dihydrate (CaCl$_2$·2H$_2$O, assay ≥99%), lithium chloride (LiCl, assay ≥99%), zinc bromide (ZnBr$_2$, assay = 99%), sodium chloride (NaCl, assay ≥99%), aluminum chloride hexahydrate (AlCl$_3$·6H$_2$O, assay ≈ 99%), CNT (multi-walled, >90% carbon basis), PVA (M$_w$ 89,000–98,000, 99 + % hydrolyzed), acrylamide (AM, assay ≥99%), and alginic acid sodium salt from brown algae (Alg, medium viscosity), Pluronic F-127, lithium fluoride (LiF, assay = 99.995%), and hydrocholoric acid aqueous solution (HCl, 35%) were purchased from Sigma-Aldrich. Laponite-RD (Nanoclay) was purchased from BYK Additives & instruments. Layered ternary carbide (Ti$_3$AlC$_2$) MAX-phase powder (particle size <200 μm) was obtained from Carbon-Ukraine Ltd. Water is Milli-Q water.

### Preparation of MXene nanosheets
MXene (Ti$_3$C$_2$T$_x$ nanosheets were synthesized based on a previous study[76]. In detail, LiF (2 g) was mixed in 40 mL of a 9 M HCl solution in a perfluoroalkoxy alkane flask for 30 min at 35 °C. A 2 g of Ti$_3$AlC$_2$ MAX-phase powder was carefully added to the LiF-HCl mixture and then mixed for 24 h at 35 °C to obtain a MXene-dispersed solution. This acidic solution was neutralized through centrifugation at 1507 × g for 5 min until the pH of the supernatant reached ~6. Afterward, a bath sonication was conducted to the neutralized MXene dispersion to delaminate the MXene. By-products and non-delaminated MXene were removed through centrifugation at 651 × g for 5 min, and then the supernatant was freeze-dried overnight to obtain a powder of delaminated multi-layered MXene nanosheets.

### Preparation of PNIPAM-based solutions and their printing
Aqueous PNIPAM solutions were prepared by mixing NIPAM monomer, APS (3 mol/mol% of NIPAM) as an initiator, and TMEDA (2 mol/mol% of NIPAM) as an accelerator for free radical polymerization. The PNIPAM concentration was controlled from 0.5 to 1.6 M. For 1.0 M PNIPAM-based composite solutions, a MXene, CNT, PVA, PAM, or Alg dispersion/solution was mixed with the NIPAM, APS, TMEDA mixture to synthesize a PNIPAM/Dye, PNIPAM/MXene, PNIPAM/CNT, PNIPAM/PVA, PNIPAM/PAM, or PNIPAM/Alg composite solution. For the PNIPAM/MXene and PNIPAM/CNT solutions, the final concentration of MXene and CNT was set as the particle/polymer wt/wt ratio was 10%, respectively. For the PNIPAM/MXene+CNT solution, the final concentration of the sum of MXene and CNT (1:1 weight ratio) was 20%. For the PNIPAM/PVA and PNIPAM/Alg solutions, the final concentration of PVA and Alg was prepared as 2.3% (wt/wt). For the PNIPAM/PAM solution, a PAM solution that was first synthesized by polymerizing AM monomer, APS (5 mol/mol% of AM), and TMEDA (4 mol/mol% of AM) was mixed with the NIPAM, APS, TMEDA mixture to obtain the PNIPAM/PAM solution comprising 1 M PAM. In the case of the PNIPAM/Dye solution, a few microliters of food dye were added to the as-prepared PNIPAM solution. The solutions were used after at least a day to ensure the polymerization with complete monomer conversion based on the previous studies[77,78]. The molecular weight (Mw) of PNIPAM and PAM were ~6.58 × 10$^6$ and 5.97 × 10$^5$ g mol$^{-1}$, respectively, according to gel permeation chromatography (size-exclusion chromatography) analysis. The 1.0 M PNIPAM-based solutions were printed into pre-designed structures using a 3D printer (ROKIT INVIVO) and a blunt needle of 20–25 gauge at room temperature (~22 °C). For spontaneous and rapid solidification of the extruded solution, 3 M CaCl$_2$ solution was mainly used unless the other was notified in the main text.

### Preparation of a support bath
Pluronic F-127 aqueous solution (35 wt/wt%) was prepared by dissolving Pluronic F-127 powder in water at an ice bath. A 6 M CaCl$_2$ aqueous solution was then blended with the prepared Pluronic F-127 solution in a 1:1 volume ratio to obtain the mixture with ~3 M CaCl$_2$, and the mixture was stored in a freezer for a full dissolution in a liquid state. This mixture was then gelated at room temperature to obtain a physical gel for a support bath.

### Preparation of PNIPAM/nanoclay bottom matrix for folding actuators
The precursor solution for PNIPAM/nanoclay bottom matrix was prepared following a previously reported method[12]. In detail, 10 mL of 2 M NIPAM monomer solution, 120 μL of 0.13 M MBAA solution as a chemical crosslinker, 0.04 g of Irgacure 2959 as an initiator, and 1 g of nanoclay as a rheological modifier were thoroughly mixed until no visible nanoclay aggregate was observed. The resultant mixture was loaded into a plastic syringe, and bubbles in the syringe were removed using the centrifuge. The mixture was printed into a pre-designed structure using a 3D printer (CELLINK BIO X) onto a flat glass substrate and then cured under 365 nm ultraviolet at 253 mW cm$^{-2}$ for 144 s (Omnicure).

### Rheological measurements
Storage modulus–temperature curves were obtained using a rheometer with a 40 mm cone plate and a 500 μm truncation gap (TA Instrument, Discovery HR-3). This rheological measurement was conducted at a frequency of 1.0 Hz, a strain of 1.0%, and a heating/cooling rate of 2 °C/min. Pure 1 M PNIPAM solution without salts was placed on the rheometer stage of ~10 °C to investigate the typical LCST of PNIPAM solution. For the LCST reduced by the salting-out effect, the 1 M PNIPAM solidified within a salt solution of 1–4 M NaCl, CaCl$_2$, or AlCl$_3$ was placed on an ~–20 °C rheometer stage, equilibrated until it transited to the coil state, and then warmed up. The LCST was determined

as the onset (starting point) of an increase in the storage modulus. For the measurement of changes in storage and loss moduli over time following the addition of CaCl$_2$ solution, the PNIPAM solution was subjected to an angular frequency of 10 rad s$^{-1}$, a strain of 1.0%, and a 1000 μm gap. The CaCl$_2$ solution diffused into the PNIPAM solution, through the edge of the PNIPAM solution placed in between the rheometer top geometry and bottom stage. In the case of the viscosity measurement over the shear rate (i.e., flow sweep) and moduli measurement over the frequency change (i.e., frequency sweep), samples were tested on a 20 °C stage with the 40 mm cone plate at a 52 μm truncation gap. During the frequency sweep, 1.0% oscillation strain was applied.

### Adhesion force measurement
To measure the adhesion force between the PNIPAM/PVA solution and the PNIPAM/PVA solution (Supplementary Fig. 8c, d), we applied 2 mL of PNIPAM/PVA solution on the rheometer bottom stage and 1 mL of PNIPAM/PVA solution on the rheometer top geometry, respectively. Afterward, two solutions were contacted for 20 s at the set gap of 1000 μm, and the tensile force was recorded, resulting from the top geometry moving upward. In the case of adhesion force between the solidified PNIPAM/PVA (as the first layer) and the PNIPAM solution (as the second layer extruded top on the first layer), we applied 2 mL of PNIPAM/PVA solution on the rheometer bottom stage and poured 3 M CaCl$_2$ solution to the PNIPAM/PVA solution. The PNIPAM/PVA solution was slightly solidified by salt ions for 20 s, and the top geometry with 1 mL of PNIPAM/PVA solution was promptly contacted with the slightly solidified PNIPAM/PVA placed on the bottom stage for 20 s at the set gap of 1000 μm. The tensile force resulting from the top geometry moving upward was recorded.

### Characterization
X-ray diffraction (XRD) analysis was performed under ambient conditions in open air, using Anton Paar XRDynamic 500. SEM images were obtained from an FEI Quanta FEG 250 SEM. Lyophilized samples frozen in liquid nitrogen and dried at −80 °C for 3 d (Labconco FreeZone) were prepared and coated with iridium for 8 s for SEM imaging. FTIR spectra data were obtained using the attenuated total reflectance (ATR) mode of a Thermo Scientific™ Nicolet™ iS50 FTIR spectrometer with a diamond ATR attachment. Conductivity was measured using a 2-probe Ohmmeter mode of Keithley 2450 multimeter. Tensile mechanical tests were performed using the Instron 5982 universal testing machine with a 100 N load cell. Filament-shaped specimens of ~1 mm diameter and rectangle-shaped specimens of ~8 mm width and ~1.5 mm thickness were prepared and tested at 5 mm min$^{-1}$ load speed. Digital optical microscope (Keyence VHX) was used to capture images of the folding actuator. Gel permeation chromatography analysis was performed using 4.0 mg/mL PNIPAM and 7.1 mg/mL PAM aqueous solutions through Shimadsu™ LC-2050 having UV detector, with Wyatt Instruments™ MALS light scattering (LS) and OptiLab differential refractive index (RI) detectors. The molecular weight of polymers was almost similar regardless of the detectors, and the molecular weight stated above was calculated based on the UV detector.

## Data availability
All data are available in the main text or the supplementary materials. Source data are provided with this paper.

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

## Acknowledgements

This research was supported by the National Science Foundation through the University of California San Diego Materials Research Science and Engineering Center (UCSD MRSEC), grant number DMR-2011924 (J.B.), and the Basic Science Research Program through the National Research Foundation of Korea (NRF) funded by the Ministry of Education, grant number RS-2023-00241263 (D.J.).

## Author contributions

Conceptualization: D.J., J.L., J.Z., J.B.; methodology: D.J., J.L., J.Z., Y.R., M.L., J.B.; investigation: D.J., J.L., J.Z., Y.R., M.L., H.S., T.H.H., J.B.; writing: D.J., J.L., J.Z., J.B.; revision: D.J. J.B.; supervision: J.B.

## Competing interests

A patent (J.B., J.L., D.J.) was filed for this work through the UCSD Office of Innovation and Commercialization. The authors declare no competing interests.
