## [Peer Review File · Nature Communications]

REVIEWER COMMENTS

Reviewer #1 (Remarks to the Author):

In "Sustainable 3D printing by reversible salting-out effects with aqueous salt solutions," the authors leverage the aggregation of pNIPAM to develop a new 3D printing method. The key innovation is the control over the LCST of pNIPAM through the ionic strength of a support bath. I believe this key idea is unique and may represent a significant development in the field if they can show that their approach produces high-quality structures. Unfortunately, the manuscript falls short in this capacity, preventing me from supporting publication in its current form.

In terms of literature review, I believe the authors need to better contextualize their method. Direct Writing of polymers into baths that drive their aggregation and assembly has been done for a long time (see the seminal work from the Lewis lab throughout the 2000's, in particular the work from Gratson and Xu. Phase separation/aggregation in embedded 3D printing was also reviewed recently in *Biophysical Reviews* by Duraivel, et. al., 2022). I believe the current work shares many aspects of previous work that leveraged polyelectrolyte/ion assembly and phase separation/aggregation of inks within support baths. While the specific idea of tuning the LCST of pNIPAM with salt seems to be unique, it is important to discuss the similarities and differences with these previously published works.

Beyond the issue of contextualizing their work in the literature, the current manuscript needs more in-depth exploration to convince the reader that the printed products are worthwhile. For example, the elastic modulus of the inks are shown as a function of temperature. Unfortunately, this does not demonstrate that the inks solidify at any of these temperatures; the authors need to show that the elastic modulus rises above the viscous modulus in these regimes. Moreover, since this ink appears to be novel, it would be good to do broader rheological characterization. What does its frequency dependence look like? Does the material look like a Maxwell fluid or a truly stable solid over a wide frequency range? How strong is the solidified ink? Some type of yielding test is needed. Moreover, single extruded filaments appear to be stable, but how robust are test structures? Single extruded filaments were tested manually (Fig 3L), but I am concerned that layer-layer adhesion will be extremely weak (as was encountered in the polyelectrolyte aggregation work for many years). Only one multi-layered structure was removed from the support materials and shown in Fig4a. Single filaments and multi-layered test specimens should be made and tested for mechanical strength. I believe all these extra rheological experiments should be very easy for the authors to conduct. They are absolutely needed in order to provide the readers confidence that the authors' approach is worthwhile.

Additionally, I am concerned about the effects the high ionic strength has on the F-127 pluronic support bath. While using F-127 has become routine in embedded 3D printing, I believe it is important to

perform a thorough rheological characterization of the modified support material in order for the readers to have confidence that this approach is not limited by changes to its rheological behaviors.

Finally, more detail is needed about the electrical conduction testing. Was the structure removed from the support before testing, or was it tested while still embedded? What was the circuit? What current and voltage? If you print the same structure without CNTs, do you get no conduction? Or just less? What level of function is needed for acceptable performance in application? As currently written, it is not clear what was done or what the importance of this demonstration is.

Overall, I believe there is potential in this manuscript and I hope the authors can make the recommended improvements to make it worthy of publication in Nature Communications.

Reviewer #2 (Remarks to the Author):

In this study, Ji et al present a novel 3D printing strategy that utilizes the reversible salting-out effects (aka. the Hofmeister series). The poly(N-isopropylacrylamide) (PNIPAm)-based inks containing various additives such as Mxene, CNT, dye, PAM, PVA etc., are capable of solidifying through a reversible physical linking when in contact with concentrated salt solutions (i.e. 2M CaCl₂, NaCl, AlCl₃). The printed 3D objects could be completely dissolved in water within 1 hour. In addition, the PNIPAm can be recovered by drying and re-used in the 3D process. The study is interesting and the manuscript is well written. However, my main concerns are as follows, raising issues about incomplete characterization and a flawed 3D printing process. Therefore, I would not recommend accepting this manuscript for publication in Nat Commun at this stage.

Main issues :

1. My main concern is whether this strategy enables the authors to fabricate real 3D objects with a specific thickness and intricate structure, akin to other 3D printing techniques. According to common knowledge, 3D printing processes can create 3D objects by solidifying 'ink' materials layer by layer. In this study, the PNIPAm solution solidified rapidly, making layer-by-layer printing impossible, at least not as demonstrated in the study data. The authors emphasize that this is the 'first' 3D printing technique based on salting-out effects. However, this is essentially a computer-controlled single-layer printing (2D drawing) technique, and it appears that there is currently no method for creating real 3D objects with intricate details.

2. It is well known that the molecular weight has a significant impact on the physical properties of polymers. As stated in the experimental section, the authors failed to characterize the molecular weight

of the PNIPAm. This omission makes it challenging for other researchers to replicate the experimental results accurately. What is the specific molecular weight of the synthesized PNIPAm? What was the concentration of NIPAM monomer when synthesizing PNIPAm in this study? What was the monomer conversion? Is it 100%? Is there any evidence of this? What is the molecular weight of the synthesized PAM?

3. Furthermore, it is perplexing why the authors did not simply use commercially available PNIPAm? Could you provide a reason for this?

4. It appears that the salts seem to be embedded in the PNIPAm matrix during solidification. It would be useful to have XRD results to confirm the crystal structure of the printed objects.

5. The incorporation of inorganic additives such as MXene or CNT is expected to improve the thermal/mechanical stability of the composites. Therefore, I recommend conducting mechanical testing and SEM measurements of these composites to verify their physical properties and the dispersion of the nanomaterials in the polymer matrix.

6. I would also recommend TGA testing of all PNIPAM-based inks to confirm their composition.

7. In the recyclability study, the authors conducted only one "print-recycle-print" cycle test. I would recommend multiple cycle tests to reveal the true potential of this technology. This is because of the possibility for fracture and fatigue of the PNIPAm chains during recycling.

Responses to Reviewers' Comments

The following is the Response to the Reviewers' Comments for the manuscript entitled "Sustainable 3D printing by reversible salting-out effects with aqueous salt solutions" (NCOMMS-23-44607) to *Nature Communications*.

Reviewer #1

General Comment: In "Sustainable 3D printing by reversible salting-out effects with aqueous salt solutions," the authors leverage the aggregation of pNIPAM to develop a new 3D printing method. The key innovation is the control over the LCST of pNIPAM through the ionic strength of a support bath. I believe this key idea is unique and may represent a significant development in the field if they can show that their approach produces high-quality structures. Unfortunately, the manuscript falls short in this capacity, preventing me from supporting publication in its current form.

Response: Thank you for your encouraging comment. We revised the manuscript to reflect all specific comments the reviewer made. The added/revised sentences and figures in the revised manuscript were highlighted in yellow.

Specific Comments

#1. In terms of literature review, I believe the authors need to better contextualize their method. Direct Writing of polymers into baths that drive their aggregation and assembly has been done for a long time (see the seminal work from the Lewis lab throughout the 2000's, in particular the work from Gratson and Xu. Phase separation/aggregation in embedded 3D printing was also reviewed recently in *Biophysical Reviews* by Duraivel, et. al., 2022). I believe the current work shares many aspects of previous work that leveraged polyelectrolyte/ion assembly and phase separation/aggregation of inks within support baths. While the specific idea of tuning the LCST of pNIPAM with salt seems to be unique, it is important to discuss the similarities and differences with these previously published works.

Response: Thank you for the comment. To address the reviewer's concern, we additionally described the printing techniques of aqueous polymeric inks. From a similar perspective to our work utilizing the salting-out effect for immediate solidification of extruded inks, we discussed other methods of solidifying the extruded inks with the formation of supramolecular interactions, polyelectrolyte complexes, or ionic crosslinks upon contact with a medium, including the work published in *Nature* from the Lewis lab (Gratson, G. M., Xu, M. & Lewis, J. A. Direct Writing of Three-Dimensional Webs. *Nature* 2004, 428, 386–386). and the review published in *Biophysical Reviews* by Duraivel (Duraivel, S. et al. Leveraging Ultra-Low Interfacial Tension and Liquid–Liquid Phase Separation in Embedded 3D Bioprinting. *Biophysics Rev.* 2022, 3, 031307). We also described the differences between our work and the previous works in the Discussion and Conclusion section of the revised manuscript, as follows.

(Revised manuscript, Page 13) This study demonstrated that immediate aggregation of PNIPAM chains upon contact with a salt solution allowed extrusion-based 3D printing. From the perspective of interchain bonding formation, previous studies also showcased the solidification of extruded inks due to the formation of supramolecular interactions^{69,70}, polyelectrolyte complexes^{71,72}, or ionic crosslinks⁷³, upon contact with a certain medium. Such interchain interactions were formed in the extruded ink or between the ink and medium^{74,75}. Note that we here excluded discussion on typical methods requiring post-processing steps for solidification, such as chemical crosslinking under heat or light for solidification. In particular, the formation of polyelectrolyte complexes between polyelectrolyte chains with different anionic and cationic groups upon contact with water or alcohol-water mixture can be analogous to the mechanism responsible for the formation of intermolecular hydrophobic interactions in the globule state of PNIPAM by salt ions^{69,70}. In contrast to the previous studies, the distinctiveness of the salting-out-based solidification strategy lies in its applicability to various PNIPAM-based composite solutions with functional particles (i.e., hydrophilic and/or hydrophobic additives) or polymeric materials. The functional particles and polymers were well mixed with the PNIPAM solution, and the PNIPAM-based composite solutions exhibited proper rheological properties and good printability. Moreover, the salting-out-based solidification was applicable to implement embedded 3D printing^{74,75}. The printing of PNIPAM-based solution ink inside a support bath with dissolved salt ions demonstrated the fabrication of the stable 3D solid structure.

(Added references)

- 69 Highley, C. B., Rodell, C. B. & Burdick, J. A. Direct 3D Printing of Shear-Thinning Hydrogels into Self-Healing Hydrogels. *Adv. Mater.* 27, 5075–5079, doi: 10.1002/adma.201501234 (2015).
- 70 Loebel, C., Rodell, C. B., Chen, M. H. & Burdick, J. A. Shear-Thinning and Self-Healing Hydrogels as Injectable Therapeutics and for 3D-Printing. *Nat. Protocols* 12, 1521–1541, doi:10.1038/nprot.2017.053 (2017).
- 71 Zhu, F. et al. 3D Printing of Ultratough Polyion Complex Hydrogels. *ACS Appl. Mater. Interfaces* 8, 31304–31310, doi:10.1021/acsami.6b09881 (2016).
- 72 Gratson, G. M., Xu, M. & Lewis, J. A. Direct Writing of Three-Dimensional Webs. *Nature* 428, 386–386, doi: 10.1038/428386a (2004).
- 73 Palma, J. H., Bertuola, M. & Hermida, É. B. Modeling Calcium Diffusion and Crosslinking Dynamics in a Thermogelling Alginate-Gelatin-Hyaluronic acid ink: 3D Bioprinting Applications. *Bioprinting* 38, e00329, doi: 10.1016/j.bprint.2024.e00329 (2024).
- 74 Duraivel, S. et al. Leveraging Ultra-Low Interfacial Tension and Liquid–Liquid Phase Separation in Embedded 3D Bioprinting. *Biophysics Rev.* 3, 031307 doi:10.1063/5.0087387 (2022).
- 75 Shiowski, D. J., Hudson, A. R., Tashman, J. W. & Feinberg, A. W. Emergence of FRESH 3D Printing as a Platform for Advanced Tissue Biofabrication. *APL Bioeng.* 5, doi:10.1063/5.0032777 (2021).

#2. Beyond the issue of contextualizing their work in the literature, the current manuscript needs more in-depth exploration to convince the reader that the printed products are worthwhile.

Response: We thank the reviewer for this comment. According to the reviewer’s suggestion, we demonstrated printing more fine and complex structures using small-diameter nozzles of 200–600 μm (Fig. 5c, and Supplementary Fig. S7 and S8) in the revised manuscript.

Revised Fig. 5c. (i) Side and (ii) top view photographs displaying two different solidified PNIPAM/PVA structures with different sizes printed in the middle of the bath.

Supplementary Fig. S7. Finely printed structures of PNIPAM/CNT composite using different nozzle sizes. The structure was printed using (a) a 0.6 mm-diameter nozzle and (b) a 0.25 mm-diameter nozzle, respectively. This fine and repetitive structure was consistently printed on a cm-scale ($4 \times 4 \text{ cm}^2$).

Supplementary Fig. S8. Printing of PNIPAM/PVA composite solution. (a) Photographs showing multi-layer printing of PNIPAM/PVA solution ink with red dye using a 200- μm -diameter nozzle. (b) Photograph of ten-layer printed grid structure of PNIPAM/PVA.

(Revised manuscript, page 10) The PNIPAM solutions were applied as 3D printing inks, without rheological modifiers, chemical crosslinkers, or additional post-processing steps/equipment that are typically required for 3D printing (Fig. 5). The PNIPAM-based solution inks were printed on a salt-solution-wetted glass substrate and instantly solidified (Fig. 5a, i). For example, the resulting printed structure of PNIPAM/CNT remained intact without the need for any post-processing or curing steps (Fig. 5a and Supplementary Fig. S7). Although we managed to print a honeycomb-patterned structure with approximately ten layers using a $\sim 200\text{-}\mu\text{m}$ -diameter nozzle (Fig. 5a, ii), weak layer-layer adhesion between the globule-state-aggregated PNIPAM in each layer and intrinsic mechanical limitation of PNIPAM produced mechanically weak solid structures. This could be addressed by using

polymer composite solution ink (Supplementary Fig. S8), such as PNIPAM/PVA forming the entanglement in polymer chains along with intrinsically good mechanical properties of PVA (Fig. 3n–q). The PNIPAM/PVA solution ink formed firm layer-layer adhesion between the solidified bottom layer containing salt ions and the freshly-printed solution top layer (Supplementary Fig. S8a–d).

(Revised manuscript, page 11) The extruded solution inks were effectively immobilized and solidified in the middle of the support bath (Fig. 5b and Supplementary Fig. S9 and Movie S1). A physical gel serving as the support bath was prepared using an aqueous blend of Pluronic F127 and CaCl₂ (Supplementary Fig. S10). For better visualization, the PNIPAM/CNT solution ink was extruded and immediately solidified upon contact with salt ions dissolved in the support bath; thereby, the printed structure was preserved in the middle of the bath. The PNIPAM/CNT structure that was solidified vertically in a spiral shape could be pulled out from the bath while maintaining its structural stability without failure (Supplementary Fig. S11a). The PNIPAM/PVA solution ink was also successfully printed and solidified in the middle of the bath at vertical and/or horizontal nozzle movements (Fig. 5c) and could be pulled out from the bath without fracture (Supplementary Fig. S11b). Air bubbles in the supporting bath surrounding printed structures were formed probably due to the rapid movement of the printing nozzle. These would be minimized through further control of the printing parameter and the rheology of the supporting bath.

#3. The elastic modulus of the inks are shown as a function of temperature. Unfortunately, this does not demonstrate that the inks solidify at any of these temperatures; the authors need to show that the elastic modulus rises above the viscous modulus in these regimes.

Response: We agree with the reviewer's comment. To clarify whether the salting-out effect on the PNIPAM solution is employable for printing techniques, we investigated whether the extruding PNIPAM solution can quickly solidify by the salt ions at room temperature (20–25 °C) adapted for general printing environments. For that, we examined the storage and loss moduli changes of the PNIPAM solution upon contact with salt ions. To address the reviewer's comment, we added a new figure showing the changes in both storage and loss moduli of the PNIPAM solution upon contact with a CaCl₂ solution with different concentrations (Fig. 3b). Also, the experimental procedure was explained in the revised manuscript.

Revised Fig. 3b. Storage (G') and loss moduli (G'') changes of PNIPAM solution by adding 2–4 M CaCl_2 solutions.

(*Revised manuscript, pages 7–8*) To validate this spontaneous and instant solidification, we examined the storage (G') and loss (G'') moduli changes of the 1M PNIPAM solution upon contact with different concentrations of CaCl_2 solution (Fig. 3b). Before the solidification, the PNIPAM solution exhibited G'' slightly larger than G' . Upon the addition of CaCl_2 solution to the PNIPAM solution at 120 s, G' significantly increased, surpassing G'' , resulting from the phase transition and solidification of PNIPAM. The rate of changes in G' was faster with a higher concentration of CaCl_2 solution.

(*Revised manuscript, Experimental section, page 16*) For the measurement of changes in storage and loss moduli over time following the addition of CaCl_2 solution, the PNIPAM solution was subjected to an angular frequency of 10 rad s^{-1} , a strain of 1.0 %, and a $1,000 \text{ } \mu\text{m}$ gap. The CaCl_2 solution diffused into the PNIPAM solution, through the edge of the PNIPAM solution placed in between the rheometer top geometry and bottom stage.

#4. Moreover, since this ink appears to be novel, it would be good to do broader rheological characterization. What does its frequency dependence look like? Does the material look like a Maxwell fluid or a truly stable solid over a wide frequency range?

Response: Thank you for the comment. According to the reviewer's suggestion, we included additional rheological characterization of PNIPAM-based solution inks in the revised manuscript (Fig. 4d–f and Supplementary Fig. S8g). Fig. 4 demonstrates the shear-thinning property (viscosity as a function of shear rate) and the viscoelastic fluidic property (shear moduli as a function of frequency) of each solution ink, along with the corresponding description in the revised manuscript as follows.

Fig. 4. Rheological characteristics of PNIPAM-based solutions. Viscosity as a function of shear rate for **a** pure PNIPAM solutions with different concentrations and **b**, **c** 1.0 M PNIPAM-based composite solutions containing functional materials or other polymers, respectively. G' and G'' as a function of angular frequency for **d** pure PNIPAM solutions with different concentrations and **e**, **f** 1.0 M PNIPAM-based composite solutions, respectively. Error bars correspond to standard deviations at $n \geq 3$.

(Revised manuscript, page 10) The frequency sweep confirmed that PNIPAM solutions were a viscoelastic fluid overall, rather than a stable solid (Fig. 4d)^{15,57}. The composite solutions also exhibited the feature of viscoelastic fluid with slightly increased moduli than those of the pure PNIPAM solution (Fig. 4d–f). This result suggests that the PNIPAM viscoelastic solutions (i.e., 1 M and above) with appropriate shear moduli and viscosity and shear-thinning behavior can easily be printed without requiring a high-extruding force. Although these solutions were in a state of liquid with $G'' > G'$ unlike typical solid-state printing inks with $G' > G''$ (ref.15,57), the spontaneous and instant solidification of PNIPAM-based solutions by the salting-out effect (Fig. 3) allowed us to implement 3D printing without additional rheological modifiers, chemical crosslinkers, post-processing steps, and specialized equipment.

(Revised manuscript, Experimental section, page 16) In the case of the viscosity measurement over the shear rate (i.e., flow sweep) and moduli measurement over the frequency change (i.e., frequency sweep), samples were tested on a 20 °C stage with the 40 mm cone plate at a 52 μm truncation gap. During the frequency sweep, 1.0 % oscillation strain was applied.

In addition, we confirmed that the PNIPAM/PVA polymer composite solution exhibits a shear-yielding property by measuring storage and loss moduli as a function of oscillation shear stress in the revised manuscript as follows.

Supplementary Fig. S8g. G' and G'' as a function of oscillation shear stress, demonstrating shear-yielding property of PNIPAM/PVA solution.

(Revised manuscript, page 11) Because the PNIPAM-based solution inks exhibited the feature of low viscosity (10–100 Pa·s at a shear rate of 0.1 s^{-1} , the state of almost no shear force, Fig 4a–c) and $G' < G''$ (Fig 4d–f and Supplementary Fig. S8g) unlike conventional printing inks whose viscosity and G' are in a few kPa–MPa·s and larger than G'' , respectively,^{2,15,16} the PNIPAM-based solution inks in extrusion-based 3D printing can be likely suitable for fabricating relatively low-height structures rather than a few cm-scale-height objects^{58–60}.

However, the prompt solidification by the salting-out effect allowed for an easy combination with the embedded 3D printing technique that has been known to enable freeform printing omnidirectionally and the printing of complex 3D structures^{61,62}. The extruded solution inks were effectively immobilized and solidified in the middle of the support bath (Fig. 5b and Supplementary Fig. S9 and Movie S1).

(Added references)

- 58 Liu, Y. et al. Soft and Elastic Hydrogel-Based Microelectronics for Localized Low-Voltage Neuromodulation. *Nat. Biomed. Eng.* 3, 58–68, doi: 10.1038/s41551-018-0335-6 (2019).
- 59 Tringides, C. M. et al. Viscoelastic Surface Electrode Arrays to Interface with Viscoelastic Tissues. *Nat. Nanotechnol.* 16, 1019–1029, doi: 10.1038/s41565-021-00926-z (2021).
- 60 Tay, R. Y., Song, Y., Yao, D. R. & Gao, W. Direct-Ink-writing 3D-Printed Bioelectronics. *Mater. Today* 71, 135–151, doi: 10.1016/j.mattod.2023.09.006 (2023).
- 62 Honaryar, H., Amirfattahi, S. & Niroobakhsh, Z. Associative Liquid-In-Liquid 3D Printing Techniques for Freeform Fabrication of Soft Matter. *Small* 19, 2206524, doi: 10.1002/smll.202206524 (2023).

#5. How strong is the solidified ink? Some type of yielding test is needed. Moreover, single extruded filaments appear to be stable, but how robust are test structures? Single extruded filaments were tested manually (Fig 3L), but I am concerned that layer-layer adhesion will be extremely weak (as was encountered in the polyelectrolyte aggregation work for many years). Only one multi-layered structure was removed from the support materials and shown in Fig4a.

Single filaments and multi-layered test specimens should be made and tested for mechanical strength. I believe all these extra rheological experiments should be very easy for the authors to conduct. They are absolutely needed in order to provide the readers confidence that the authors' approach is worthwhile.

Response: We thank the reviewer for raising important questions regarding the mechanical properties of solidified structures. As per the reviewer's suggestions, we first investigated the mechanical properties of the solidified structures in a single filament form (Fig. 3p and 3q).

Revised Fig. 3. **n** Solidified PNIPAM/PVA composite showing deformable and stretchable behaviors. **o** Schematic illustrating the entangled polymer chains. **p** Tensile stress-strain curve and **q** the corresponding tensile strength and elastic modulus of the solidified PNIPAM/PVA in a filament form (n=5). Error bars correspond to standard deviations.

(Revised manuscript, page 9) In the case of the polymer composite, for instance, the solidified PNIPAM/PVA was deformable and stretchable (Fig. 3n), which contrasted with the rigid and non-stretchable structure consisting of the PNIPAM single component. This behavior was likely attributed to the entanglement of PNIPAM and PVA chains and the intrinsically good mechanical properties of PVA (Fig. 3o)⁵²⁻⁵⁴. Indeed, the solidified PNIPAM/PVA in a filament form exhibited good mechanical properties with tensile strength and elastic modulus in the magnitude of 1 and 100 MPa, respectively (Fig. 3p and 3q).

(Revised manuscript, Experimental section, page 17) Tensile mechanical tests were performed using the Instron 5982 universal testing machine with a 100 N load cell. Filament-shaped specimens of ~1 mm diameter and rectangle-shaped specimens of ~8 mm width and ~1.5 mm thickness were prepared and tested at 5 mm min⁻¹ load speed.

(Added references)

- 52 Li, Y., Li, S. & Sun, J. Degradable Poly(vinyl alcohol)-Based Supramolecular Plastics with High Mechanical Strength in a Watery Environment. *Adv. Mater.* 33, 2007371, doi: 10.1002/adma.202007371 (2021).
- 53 Hua, M. & He, X. Soft-Fiber-Reinforced Tough and Fatigue Resistant Hydrogels. *Matter* 4, 1755–1757, doi: 10.1016/j.matt.2021.05.006 (2021).
- 54 Li, G. et al. Highly Conducting and Stretchable Double-Network Hydrogel for Soft Bioelectronics. *Adv. Mater.* 34, 2200261, doi: 10.1002/adma.202200261 (2022).

Further, as the reviewer pointed out, the pure PNIPAM solution and its solidified layer exhibited weak layer-layer adhesion, presumably because hydrophobic aggregation of

PNIPAM chains in the solidified layer could have limited interactions with the freshly extruded PNIPAM aqueous solution layer. While addressing this issue, we found that polymer composite solution (e.g., PNIPAM/PVA) could achieve relatively strong layer-layer adhesion, likely attributable to the physical crosslinks and entanglements of polymer chains across the layers during the diffusion of salt ions from the first solidified layer to the second freshly extruded solution layer. Therefore, we successfully demonstrated multi-layered printed structures (Supplementary Fig. S8a and S8b) of PNIPAM/PVA, then evaluated the layer-layer adhesion force (Supplementary Fig. S8c and S8d) and the tensile mechanical properties of the multi-layered printed structure (Supplementary Fig. S8e and S8f) in the revised manuscript as follows.

Supplementary Fig. S8. Printing of PNIPAM/PVA composite solution. (a) Photographs showing multi-layer printing of PNIPAM/PVA solution ink with red dye using a 200- μ m-diameter nozzle. (b) Photograph of ten-layer printed grid structure of PNIPAM/PVA. (c) Schematic illustration depicting how we evaluated the adhesion force between the first and the second layers, and (d) the corresponding result. In the case of PNIPAM/PVA solution and solidified PNIPAM/PVA as the dark blue graph, diffusion of salt ions from the first layer (solidified PNIPAM/PVA) to the second layer (freshly extruded PNIPAM/PVA solution) likely formed physical crosslinks and entanglement of the polymers across both layers, thereby generating the tensional force, unlike the case of PNIPAM/PVA solutions at both (sky blue graph). The detailed measurement procedure is described in the Experimental section (Adhesion force measurement) of the main article. (e) Photograph of three-layer printed structure of PNIPAM/PVA and (f) its tensile stress-strain curve showing good physical and mechanical stability with free-standing and stretchable characteristics. (g) G' and G'' as a function of oscillation shear stress, demonstrating shear-yielding property of PNIPAM/PVA solution.

(Revised manuscript, pages 10–11) The PNIPAM solutions were applied as 3D printing inks, without rheological modifiers, chemical crosslinkers, or additional post-processing steps/equipment that are typically required for 3D printing (Fig. 5). The PNIPAM-based solution inks were printed on a salt-solution-wetted glass substrate and instantly solidified (Fig. 5a, i). For example, the resulting printed structure of PNIPAM/CNT remained intact without the need for any post-processing or curing steps (Fig. 5a and Supplementary Fig. S7).

Although we managed to print a honeycomb-patterned structure with approximately ten layers using a ~200- μm -diameter nozzle (Fig. 5a, ii), weak layer-layer adhesion between the globule-state-aggregated PNIPAM in each layer and intrinsic mechanical limitation of PNIPAM produced mechanically weak solid structures. This could be addressed by using polymer composite solution ink (Supplementary Fig. S8), such as PNIPAM/PVA forming the entanglement in polymer chains along with intrinsically good mechanical properties of PVA (Fig. 3n–q). The PNIPAM/PVA solution ink formed firm layer-layer adhesion between the solidified bottom layer containing salt ions and the freshly-printed solution top layer (Supplementary Fig. S8a–d). As a result, the multi-layered structure was free-standable and stretchable (Supplementary Fig. S8e and 8f). Because the PNIPAM-based solution inks exhibited the feature of low viscosity (10–100 Pa·s at a shear rate of 0.1 s^{-1} , the state of almost no shear force, Fig 4a–c) and $G' < G''$ (Fig 4d–f and Supplementary Fig. S8g) unlike conventional printing inks whose viscosity and G' are in a few kPa–MPa·s and larger than G'' , respectively,^{2,15,16} the PNIPAM-based solution inks in extrusion-based 3D printing can be likely suitable for fabricating relatively low-height structures rather than a few cm-scale-height objects^{58–60}.

***(Revised manuscript, Experimental section, pages 16–17)* Adhesion force measurement.**

To measure the adhesion force between the PNIPAM/PVA solution and the PNIPAM/PVA solution (Supplementary Fig. S8c and S8d), we applied 2 mL of PNIPAM/PVA solution on the rheometer bottom stage and 1 mL of PNIPAM/PVA solution on the rheometer top geometry, respectively. Afterward, two solutions were contacted for 20 s at the set gap of 1,000 μm , and the tensile force was recorded, resulting from the top geometry moving upward. In the case of adhesion force between the solidified PNIPAM/PVA (as the first layer) and the PNIPAM solution (as the second layer extruded top on the first layer), we applied 2 mL of PNIPAM/PVA solution on the rheometer bottom stage and poured 3 M CaCl_2 solution to the PNIPAM/PVA solution. The PNIPAM/PVA solution was slightly solidified by salt ions for 20 s, and the top geometry with 1 mL of PNIPAM/PVA solution was promptly contacted with the slightly solidified PNIPAM/PVA placed on the bottom stage for 20 s at the set gap of 1,000 μm . The tensile force resulting from the top geometry moving upward was recorded.

#6. Additionally, I am concerned about the effects the high ionic strength has on the F-127 Pluronic support bath. While using F-127 has become routine in embedded 3D printing, I believe it is important to perform a thorough rheological characterization of the modified support material in order for the readers to have confidence that this approach is not limited by changes to its rheological behaviors.

Response: Thank you for the comment. As the Reviewer's suggestion, we performed the rheological characterization of the modified Pluronic F-127 support bath with CaCl_2 . Therefore, in the revised manuscript, we corroborated that the support bath is solid at room temperature and can become a fluid-like state while subjected to appropriate shear stress over the yield point generated by the syringe nozzle moving inside the bath (Supplementary Fig. S10).

Supplementary Fig. S10. Rheological characteristics of Pluronic F-127 support bath containing 3 M CaCl₂. G' and G'' (a) as a function of temperature and (b) as a function of shear stress. The liquid-state mixture was prepared at low temperatures and was a stable solid at ambient temperatures. The solid support bath temporarily became in a fluid-like state while subjected to shear stress (over yield point) generated by the syringe nozzle moving inside the bath.

(Revised manuscript, page 11) The extruded solution inks were effectively immobilized and solidified in the middle of the support bath (Fig. 5b and Supplementary Fig. S9 and Movie S1). A physical gel serving as the support bath was prepared using an aqueous blend of Pluronic F127 and CaCl₂ (Supplementary Fig. S10).

(Revised manuscript, Experimental section page 16) Pluronic F-127 aqueous solution (35 wt/wt%) was prepared by dissolving Pluronic F-127 powder in water at an ice bath. A 6 M CaCl₂ aqueous solution was then blended with the prepared Pluronic F-127 solution in a 1:1 volume ratio to obtain the mixture with ~3 M CaCl₂, and the mixture was stored in a freezer for a full dissolution in a liquid state. This mixture was then gelled at room temperature to obtain a physical gel for a support bath.

#7. Finally, more detail is needed about the electrical conduction testing. Was the structure removed from the support before testing, or was it tested while still embedded? What was the circuit? What current and voltage? If you print the same structure without CNTs, do you get no conduction? Or just less? What level of function is needed for acceptable performance in application? As currently written, it is not clear what was done or what the importance of this demonstration is.

Response: We thank the reviewer for this comment. In the revised manuscript, we investigated the conductivity of the PNIPAM single-component and the PNIPAM composite containing conductivity particles (i.e., CNT and MXene) (Fig. 3m). To measure the conductivity, pure PNIPAM solution and PNIPAM/MXene+CNT solution was respectively extruded and solidified in a filament shape inside the CaCl₂ solution. Next, water on the surface of the solidified samples was gently wiped using Kimtech Wipes, and the filament-shaped solids were placed on the conductivity-measuring device (Supplementary Fig. S6). In the revised manuscript, we representatively compared the conductivity of pure PNIPAM and

PNIPAM/MXene+CNT composite (polymer/particle wt/wt ratio = 20 % and MXene:CNT = 1:1 weight ratio), further detailed studies on the effect of content of conductive particles and of the ratio between CNT and MXene particles on the enhancement of conductivity could be additionally performed in the near future.

Fig. 3m. Conductivity of pure PNIPAM and PNIPAM/MXene+CNT composite solidified by 3 M CaCl₂ (n=4). Error bars correspond to standard deviations.

Supplementary Fig. S6. Measurement setup for the conductivity of solidified PNIPAM and PNIPAM/MXene+CNT. To demonstrate the effect of conductive inorganic particles (MXene and CNT) on the conductivity enhancement, we compared pure PNIPAM and PNIPAM/MXene+CNT composite representatively. Each solution was extruded into the 3 M CaCl₂ solution, resulting in fully crosslinked and solidified samples. Water on the surface of the solidified samples was gently wiped using Kimtech Wipes, and the sample was placed on the customized measuring device as shown in the schematic illustration.

(Revised manuscript, page 9) The solidified PNIPAM structures were free-standing overall and could have enhanced mechanical properties and/or conductivity depending on the additive component. For example, the solidified PNIPAM/MXene (Fig. 3j) demonstrated that small particles were instantly entrapped among the aggregated PNIPAM chains in the globule state before diffusing out to the surroundings (Fig. 3k). This instant aggregation was shown as an immediate increase in storage modulus upon contact of the solution with salt ions, and the composite aggregates with stiff particles resulted in higher storage modulus (Fig. 3l). Further, the solidified structure containing MXene and CNT particles demonstrated a higher conductivity than that of the solidified structure of pure PNIPAM (Fig. 3m and Supplementary Fig. S6) due to the conductive nature of CNT and MXene particles⁴⁷⁻⁵¹.

(Revised manuscript, Experimental section, page 15) For the PNIPAM/MXene and PNIPAM/CNT solutions, the final concentration of MXene and CNT was set as the particle/polymer wt/wt ratio was 10 %, respectively. For the PNIPAM/MXene+CNT solution, the final concentration of the sum of MXene and CNT (1:1 weight ratio) was 20 %.

(Revised manuscript, Experimental section, page 17) Conductivity was measured using a 2-probe Ohmmeter mode of Keithley 2450 multimeter.

(Added references)

- 47 Shin, M. K. et al. Synergistic Toughening of Composite Fibres by Self-Alignment of Reduced Graphene Oxide and Carbon Nanotubes. *Nat. Commun.* 3, 650, doi: 10.1038/ncomms1661 (2012).
- 48 Jia, X. et al. Dramatic Enhancements in Toughness of Polyimide Nanocomposite via Long-CNT-Induced Long-Range Creep. *J. Mater. Chem.* 22, 7050–7056, doi: 10.1039/C2JM15359A (2012).
- 49 Eom, W. et al. Large-Scale Wet-Spinning of Highly Electroconductive MXene Fibers. *Nat. Commun.* 11, 2825, doi: 10.1038/s41467-020-16671-1 (2020).
- 50 Wan, S. et al. High-Strength Scalable MXene Films through Bridging-Induced Densification. *Science* 374, 96–99, doi: 10.1126/science.abg2026 (2021).
- 51 Ghaffarkhah, A. et al. Ultra-Flyweight Cryogels of MXene/Graphene Oxide for Electromagnetic Interference Shielding. *Adv. Funct. Mater.* 33, 2304748, doi: 10.1002/adfm.202304748 (2023).

As confirmed the rheological characteristics of the PNIPAM-based solutions (Fig. 4), we next demonstrated the printing of the solutions was implementable onto a CaCl_2 -wetted glass substrate (Revised Fig. 5a). Subsequently, the conductive characteristic of the printed structure containing conductive particles was visualized by connecting a light bulb (Fig. 5e). In this demonstration, the structure, which was printed onto the CaCl_2 -wetted substrate instead of being immersed in a salt solution or embedded in a support bath, was in a dried state to eliminate other factors affecting the electrical conductivity besides conductive particles. By comparing the operation of a light bulb that was connected to a solidified and dried PNIPAM structure (Supplementary Fig. S13a), we confirmed that the pure PNIPAM structure does not have electrical conduction (Supplementary Fig. 13b), whereas the PNIPAM/CNT composite structure possess electrical conduction (Supplementary Fig. 13c). Such results imply that the PNIPAM/CNT solution ink can be utilized for printing electrically conductive structures. In this study, we incorporated 10 % CNTs in the printed structure (details in the Experimental section), and the consequential printed structure was compatible with a 25–30 V (Supplementary Fig. 13c). Because this voltage can be provided by conventional batteries, the printed structure could be utilized as electric circuits or electrode arrays. Furthermore, we are expecting that additional studies can increase the electrical conductivity of the printed circuit/array and reduce the compatible voltage to exploit the printed structures in advanced electrical devices (e.g., bioelectronics) connecting with low-voltage batteries (e.g., a few voltages).

Revised Fig. 5a. Printing of PNIPAM-based ink solution onto a substrate wetted by a salt solution for spontaneous and rapid solidification.

Supplementary Fig. S13. Conductivity comparison between pure PNIPAM and PNIPAM/CNT composite. (a) Experimental setup. The printed and solidified PNIPAM structures (by 3 M CaCl_2), completely dried to remove the effect of ion conduction, were connected to the power supply that applied voltage across the entire circuit. The PNIPAM structures were connected to a bulb linked to the power supply. Copper wires were tightly attached to the PNIPAM structures with copper tape. (b) While the pure PNIPAM structure did not make a light bulb work (even at 30 V) due to insufficient conductivity, (c) the PNIPAM/CNT composite structure was able to power a light bulb. The brightness gradually increased with an increase in the applied voltage.

(Revised manuscript, page 12) The PNIPAM/CNT solution was applicable for printing water-soluble disposable conductive structures (Fig. 5e and Supplementary Fig. S13). The printed and dried pure PNIPAM structure failed to enable lightbulb functioning due to the lack of electrical conduction in pure PNIPAM (Supplementary Fig. S13b). In contrast, the electrical conduction through the CNTs in the PNIPAM/CNT structure, which was printed and dried, enabled a light bulb to function. Such results suggest that the PNIPAM/CNT solution ink can be used for fabricating printable electrical circuits. The brightness of this bulb gradually increased with increasing voltage from 0 to 30 V, reaching its maximum at

25–30 V (Supplementary Fig. S13c). In this demonstration, we incorporated 10 % CNTs in the printed structure, and the resulting printed structure was compatible with a 25–30 V. Further studies (e.g., with different conductive particle contents) could possibly reduce the compatible voltage of the electric-circuit/electrode-array for advanced electrical devices connecting with low-voltage batteries^{58,59,65,66}. Meanwhile, because this printed structure comprising physically crosslinked PNIPAM was water-soluble, it demonstrates promise for developing disposable yet environmentally friendly **electrically conductive structures** (Fig. 5e).

(Revised manuscript, page 14) In terms of electrically conductive structures, we could further conduct studies on the effect of the content of conductive particles (e.g., CNT and MXene particles) and on the way of obtaining homogeneous conductive PNIPAM solutions (Supplementary Fig. S17) to increase electrical conductivity for advanced electrical devices (e.g., bioelectronics).

(Added references)

- 58 Liu, Y. et al. Soft and Elastic Hydrogel-Based Microelectronics for Localized Low-Voltage Neuromodulation. *Nat. Biomed. Eng.* 3, 58–68, doi: 10.1038/s41551-018-0335-6 (2019).
- 59 Tringides, C. M. et al. Viscoelastic Surface Electrode Arrays to Interface with Viscoelastic Tissues. *Nat. Nanotechnol.* 16, 1019–1029, doi: 10.1038/s41565-021-00926-z (2021).
- 65 Hinchet, R. et al. Transcutaneous Ultrasound Energy Harvesting using Capacitive Triboelectric Technology. *Science* 365, 491–494, doi: 10.1126/science.aan3997 (2019).
- 66 Zhao, Z., Spyropoulos, G. D., Cea, C., Gelinas, J. N. & Khodagholy, D. Ionic Communication for Implantable Bioelectronics. *Sci. Adv.* 8, eabm7851, doi:10.1126/sciadv.abm7851 (2022).

Reviewer #2

General Comment: In this study, Ji et al present a novel 3D printing strategy that utilizes the reversible salting-out effects (aka. the Hofmeister series). The poly(N-isopropylacrylamide) (PNIPAm)-based inks containing various additives such as Mxene, CNT, dye, PAM, PVA etc., are capable of solidifying through a reversible physical linking when in contact with concentrated salt solutions (i.e. 2M CaCl₂, NaCl, AlCl₃). The printed 3D objects could be completely dissolved in water within 1 hour. In addition, the PNIPAm can be recovered by drying and re-used in the 3D process. The study is interesting and the manuscript is well written. However, my main concerns are as follows, raising issues about incomplete characterization and a flawed 3D printing process. Therefore, I would not recommend accepting this manuscript for publication in Nat Commun at this stage.

Response: Thank you for your encouraging comment. We revised the manuscript to reflect all specific comments the reviewer made. The added/revised sentences and figures in the revised manuscript were highlighted in yellow.

Specific Comments

#1. My main concern is whether this strategy enables the authors to fabricate real 3D objects with a specific thickness and intricate structure, akin to other 3D printing techniques.

According to common knowledge, 3D printing processes can create 3D objects by solidifying 'ink' materials layer by layer. In this study, the PNIPAm solution solidified rapidly, making layer-by-layer printing impossible, at least not as demonstrated in the study data. The authors emphasize that this is the 'first' 3D printing technique based on salting-out effects. However, this is essentially a computer-controlled single-layer printing (2D drawing) technique, and it appears that there is currently no method for creating real 3D objects with intricate details.

Response: Thank you for the comment. In the revised manuscript, we additionally presented finely printed complex structures of PNIPAM/CNT using small-diameter nozzles (Supplementary Fig. S7) and multi-layered 3D structures of PNPAM/PVA (Supplementary Fig. S8a and S8b). As the reviewer pointed out, the pure PNIPAM solution and its solidified layer (single-component PNIPAM matrix) had relatively weak layer-layer adhesion, presumably because hydrophobic aggregation of PNIPAM chains in the solidified layer had limited interactions with the freshly extruded PNIPAM aqueous solution layer. While addressing this issue, we found that polymer composite solution (e.g., PNIPAM/PVA) managed to form relatively firm layer-layer adhesion (Supplementary Fig. S8c and S8d), likely attributable to the physical crosslinks and entanglements of polymer chains across the layers during the diffusion of salt ions from the first solidified layer to the second freshly extruded solution layer. The resulting multi-layered structure demonstrated its good physical/mechanical stability (Supplementary Fig. S8e and S8f). Furthermore, we clarified that the embedded 3D printing technique employing a support bath allows for fabricating 3D objects with both vertical and horizontal nozzle movements (Revised Fig. 5b and 5c; and Supplementary Fig. S9 and S11) in the revised manuscript.

Revised Fig. 5. b Embedded printing of PNIPAM-based solution ink in a support bath comprising Pluronic F-127 and CaCl₂. **c** (i) Side and (ii) top view photographs displaying two different solidified PNIPAM/PVA structures with different sizes printed in the middle of the bath.

Supplementary Fig. S7. Finely printed structures of PNIPAM/CNT composite using different nozzle sizes. The structure was printed using (a) a 0.6 mm-diameter nozzle and (b) a 0.25 mm-diameter nozzle, respectively. This fine and repetitive structure was consistently printed on a cm-scale ($4 \times 4 \text{ cm}^2$).

Supplementary Fig. S8. Printing of PNIPAM/PVA composite solution. (a) Photographs showing multi-layer printing of PNIPAM/PVA solution ink with red dye using a 200- μm -diameter nozzle. (b) Photograph of ten-layer printed grid structure of PNIPAM/PVA. (c) Schematic illustration depicting how we evaluated the adhesion force between the first and the second layers, and (d) the corresponding result. In the case of PNIPAM/PVA solution and solidified PNIPAM/PVA as the dark blue graph, diffusion of salt ions from the first layer (solidified PNIPAM/PVA) to the second layer (freshly extruded PNIPAM/PVA solution) likely formed physical crosslinks and entanglement of the polymers across both layers, thereby generating the tensional force, unlike the case of PNIPAM/PVA solutions at both (sky blue graph). The detailed measurement procedure is described in the Experimental section (Adhesion force measurement) of the main article. (e) Photograph of three-layer printed structure of PNIPAM/PVA and (f) its tensile stress-strain curve showing good physical and mechanical stability with free-standing and stretchable characteristics. (g) G' and G'' as a function of oscillation shear stress, demonstrating shear-yielding property of PNIPAM/PVA solution.

Supplementary Fig. S9. PNIPAM/CNT structure in the support bath. The printed solution immediately solidified upon contact with the salt ions dissolved in the support bath.

Supplementary Fig. S11. Physically/mechanically stable structures solidified in the support bath. (a) PNIPAM/CNT and (b) PNIPAM/PVA solutions were stably solidified in the bath during vertical and horizontal nozzle movements. The solidified sample was able to be pulled out from the support bath. The fast movement of the printing nozzle in the bath often made some air bubbles.

(Revised manuscript, pages 10–11) The PNIPAM solutions were applied as 3D printing inks, without rheological modifiers, chemical crosslinkers, or additional post-processing steps/equipment that are typically required for 3D printing (Fig. 5). The PNIPAM-based solution inks were printed on a salt-solution-wetted glass substrate and instantly solidified (Fig. 5a, i). For example, the resulting printed structure of PNIPAM/CNT remained intact without the need for any post-processing or curing steps (Fig. 5a and Supplementary Fig. S7). Although we managed to print a honeycomb-patterned structure with approximately ten layers using a ~200- μm -diameter nozzle (Fig. 5a, ii), weak layer-layer adhesion between the globule-state-aggregated PNIPAM in each layer and intrinsic mechanical limitation of PNIPAM produced mechanically weak solid structures. This could be addressed by using polymer composite solution ink (Supplementary Fig. S8), such as PNIPAM/PVA forming the entanglement in polymer chains along with intrinsically good mechanical properties of PVA (Fig. 3n–q). The PNIPAM/PVA solution ink formed firm layer-layer adhesion between the solidified bottom layer containing salt ions and the freshly-printed solution top layer (Supplementary Fig. S8a–d). As a result, the multi-layered structure was free-standing and stretchable (Supplementary Fig. S8e and 8f). Because the PNIPAM-based solution inks exhibited the feature of low viscosity (10–100 Pa·s at a shear rate of 0.1 s^{-1} , the state of almost no shear force, Fig 4a–c) and $G' < G''$ (Fig 4d–f and Supplementary Fig. S8g) unlike conventional printing inks whose viscosity and G' are in a few kPa–MPa·s and larger than G'' , respectively,^{2,15,16} the PNIPAM-based solution inks in extrusion-based 3D printing can be likely suitable for fabricating relatively low-height structures rather than a few cm-scale-height objects^{58–60}.

However, the prompt solidification by the salting-out effect allowed for an easy combination with the embedded 3D printing technique that has been known to enable freeform printing omnidirectionally and the printing of complex 3D structures^{61,62}. The

extruded solution inks were effectively immobilized and solidified in the middle of the support bath (Fig. 5b and Supplementary Fig. S9 and Movie S1). A physical gel serving as the support bath was prepared using an aqueous blend of Pluronic F127 and CaCl₂ (Supplementary Fig. S10). For better visualization, the PNIPAM/CNT solution ink was extruded and immediately solidified upon contact with salt ions dissolved in the support bath; thereby, the printed structure was preserved in the middle of the bath. The PNIPAM/CNT structure that was solidified vertically in a spiral shape could be pulled out from the bath while maintaining its structural stability without failure (Supplementary Fig. S11a). The PNIPAM/PVA solution ink was also successfully printed and solidified in the middle of the bath at vertical and/or horizontal nozzle movements (Fig. 5c) and could be pulled out from the bath without fracture (Supplementary Fig. S11b). Air bubbles in the supporting bath surrounding printed structures were formed probably due to the rapid movement of the printing nozzle. These would be minimized through further control of the printing parameter and the rheology of the supporting bath.

(Revised manuscript, Experimental section, pages 16–17) **Adhesion force measurement.**

To measure the adhesion force between the PNIPAM/PVA solution and the PNIPAM/PVA solution (Supplementary Fig. S8c and S8d), we applied 2 mL of PNIPAM/PVA solution on the rheometer bottom stage and 1 mL of PNIPAM/PVA solution on the rheometer top geometry, respectively. Afterward, two solutions were contacted for 20 s at the set gap of 1,000 μm, and the tensile force was recorded, resulting from the top geometry moving upward. In the case of adhesion force between the solidified PNIPAM/PVA (as the first layer) and the PNIPAM solution (as the second layer extruded top on the first layer), we applied 2 mL of PNIPAM/PVA solution on the rheometer bottom stage and poured 3 M CaCl₂ solution to the PNIPAM/PVA solution. The PNIPAM/PVA solution was slightly solidified by salt ions for 20 s, and the top geometry with 1 mL of PNIPAM/PVA solution was promptly contacted with the slightly solidified PNIPAM/PVA placed on the bottom stage for 20 s at the set gap of 1,000 μm. The tensile force resulting from the top geometry moving upward was recorded.

(Added references)

- 58 Liu, Y. et al. Soft and Elastic Hydrogel-Based Microelectronics for Localized Low-Voltage Neuromodulation. *Nat. Biomed. Eng.* 3, 58–68, doi: 10.1038/s41551-018-0335-6 (2019).
- 59 Tringides, C. M. et al. Viscoelastic Surface Electrode Arrays to Interface with Viscoelastic Tissues. *Nat. Nanotechnol.* 16, 1019–1029, doi: 10.1038/s41565-021-00926-z (2021).
- 60 Tay, R. Y., Song, Y., Yao, D. R. & Gao, W. Direct-Ink-writing 3D-Printed Bioelectronics. *Mater. Today* 71, 135–151, doi: 10.1016/j.mattod.2023.09.006 (2023).
- 62 Honaryar, H., Amirfattahi, S. & Niroobakhsh, Z. Associative Liquid-In-Liquid 3D Printing Techniques for Freeform Fabrication of Soft Matter. *Small* 19, 2206524, doi: 10.1002/sml.202206524 (2023).
- 74 Duraivel, S. et al. Leveraging Ultra-Low Interfacial Tension and Liquid–Liquid Phase Separation in Embedded 3D Bioprinting. *Biophysics Rev.* 3, doi:10.1063/5.0087387 (2022).
- 75 Shiowski, D. J., Hudson, A. R., Tashman, J. W. & Feinberg, A. W. Emergence of FRESH 3D Printing as a Platform for Advanced Tissue Biofabrication. *APL Bioeng.* 5, doi:10.1063/5.0032777 (2021).

#2. It is perplexing why the authors did not simply use commercially available PNIPAm? Could you provide a reason for this?

Response: Yes, the main reason for using NIPAM monomers is that they are much more economical than commercial PNIPAM. Specifically, PNIPAM polymer powder is “\$821.00 for 10 g” purchasable from Sigma-Aldrich, whereas the NIPAM monomer powder is only “\$234.00 for 500 g” purchasable from Tokyo Chemical Industry (TCI). An initiator for NIPAM polymerization, such as ammonium persulfate we used in this study, is even cheaper than NIPAM monomer.

#3. It is well known that the molecular weight has a significant impact on the physical properties of polymers. As stated in the experimental section, the authors failed to characterize the molecular weight of the PNIPAM. This omission makes it challenging for other researchers to replicate the experimental results accurately. What is the specific molecular weight of the synthesized PNIPAM? What was the concentration of NIPAM monomer when synthesizing PNIPAM in this study? What was the monomer conversion? Is it 100%? Is there any evidence of this? What is the molecular weight of the synthesized PAM?

Response: We agree that providing clear experimental procedures is important to enable other researchers to replicate the printing inks. According to the Reviewer’s comment, we performed gel permeation chromatography (GPC) analysis to characterize the molecular weight of polymer chains (PNIPAM of 6.58×10^6 g mol⁻¹ and PAM of 5.97×10^5 g mol⁻¹) and further revised the experimental section to clarify the specific amount of each component in the experimental section.

(Revised manuscript, Experimental section, page 15) Aqueous PNIPAM solutions were prepared by mixing NIPAM monomer, APS (3 mol/mol% of NIPAM) as an initiator, and TMEDA (2 mol/mol% of NIPAM) as an accelerator for free radical polymerization. **The PNIPAM concentration was controlled from 0.5 to 1.6 M. For the preparation of 1.0 M PNIPAM-based composite solutions,** a MXene, CNT, PVA, PAM, or Alg dispersion/solution was mixed with the NIPAM, APS, TMEDA mixture to synthesize a PNIPAM/MXene, PNIPAM/CNT, PNIPAM/PVA, PNIPAM/PAM, or PNIPAM/Alg composite solution, respectively. For the PNIPAM/MXene and PNIPAM/CNT solutions, the final concentration of MXene and CNT was set as the particle/polymer wt/wt ratio was 10 %, respectively. **For the PNIPAM/MXene+CNT solution, the final concentration of the sum of MXene and CNT (1:1 weight ratio) was 20 %.** For the PNIPAM/PVA and PNIPAM/Alg solutions, the final concentration of PVA and Alg was prepared as 2.3 % (wt/wt). For the PNIPAM/PAM solution, a PAM solution that was first synthesized by polymerizing AM monomer, APS (5 mol/mol% of AM), and TMEDA (4 mol/mol% of AM) was mixed with the NIPAM, APS, TMEDA mixture to obtain the PNIPAM/PAM solution comprising 1 M PAM. In the case of the PNIPAM/Dye solution, a few microliters of food dye were added to the as-prepared PNIPAM solution.

(Revised manuscript, Experimental section, page 15) **The molecular weight (Mw) of PNIPAM and PAM were approximately 6.58×10^6 and 5.97×10^5 g mol⁻¹, respectively, according to gel permeation chromatography (size-exclusion chromatography) analysis.**

(Revised manuscript, Experimental section, page 17) Gel permeation chromatography analysis was performed using 4.0 mg/mL PNIPAM and 7.1 mg/mL PAM aqueous solutions through Shimadzu™ LC-2050 having UV detector, with Wyatt Instruments™ MALS light scattering (LS) and OptiLab differential refractive index (RI) detectors. The molecular weight of polymers was almost similar regardless of the detectors, and the molecular weight stated above was calculated based on the UV detector.

For the PNIPAM and PAM polymerization, we added an abundant amount of initiator (3–5 mol/mol % of monomer, corresponding to 6–20 wt/wt% of monomer) for fast and conclusive polymerization. According to the previous works,^[R1–R8] the initiator amount we used in this study was more than sufficient to achieve ~100% monomer conversion. We believe the additional experimental details (the exact number of each component amounts in mol/mol % and wt/wt %) would be helpful for researchers to prepare the printing inks.

- [R1] Polymerization kinetics of polyacrylamide gels I. Effect of different cross-linkers, *ELECTROPHORESIS*, 2, 213 (1981).
- [R2] Poly(N-isopropylacrylamide): experiment, theory and application, *Prog. Polym. Sci.* 17, 163 (1992).
- [R3] High-Conversion Free-Radical Bulk Polymerization of Styrene: Termination Kinetics Studied by Electron Spin Resonance, Fourier Transform Near-Infrared Spectroscopy, and Gel Permeation Chromatography, *Macromolecules* 34, 7686–7691(2001).
- [R4] Effects of Tacticity and Molecular Weight of Poly(N-isopropylacrylamide) on Its Glass Transition Temperature, *Macromolecules* 44, 5822 (2011).
- [R5] What Does Conversion Mean in Polymer Science?, *Macromol. Chem. Phys.* 222, 2100010 (2021).
- [R6] Effects of synthesis-solvent polarity on the physicochemical and rheological properties of poly(N-isopropylacrylamide) (PNIPAm) hydrogels *J. Mater. Res. Tech.* 13, 769 (2021).
- [R7] A greener route for smart PNIPAm microgel synthesis using a bio-based synthesis-solvent, *Eur. Polym. J.* 174, 111311 (2022).
- [R8] Rapid RAFT Polymerization of Acrylamide with High Conversion, *Molecules* 28, 2588 (2023).

(Revised manuscript, Experimental section, page 15) Aqueous PNIPAM solutions were prepared by mixing NIPAM monomer, APS (3 mol/mol% of NIPAM) as an initiator, and TMEDA (2 mol/mol% of NIPAM) as an accelerator for free radical polymerization. (...) For the PNIPAM/PVA and PNIPAM/Alg solutions, the final concentration of PVA and Alg was prepared as 2.3 % (wt/wt). For the PNIPAM/PAM solution, a PAM solution that was first synthesized by polymerizing AM monomer, APS (5 mol/mol% of AM), and TMEDA (4 mol/mol% of AM) was mixed with the NIPAM, APS, TMEDA mixture to obtain the PNIPAM/PAM solution comprising 1 M PAM. In the case of the PNIPAM/Dye solution, a few microliters of food dye were added to the as-prepared PNIPAM solution. The solutions were used after at least a day to ensure the polymerization with complete monomer conversion based on the previous studies^{77,78}.

(Added references)

77 Gelfi, C. & Righetti, P. G. Polymerization Kinetics of Polyacrylamide Gels I. Effect of Different Cross-linkers. *ELECTROPHORESIS* 2, 213–219, doi: 10.1002/elps.1150020404 (1981).

78 Biswas, C. S. et al. Effects of Tacticity and Molecular Weight of Poly(N-isopropylacrylamide) on Its Glass Transition Temperature. *Macromolecules* 44, 5822–5824, doi: 10.1021/ma200735k (2011).

#4. It appears that the salts seem to be embedded in the PNIPAm matrix during solidification. It would be useful to have XRD results to confirm the crystal structure of the printed objects.

Response: We thank the reviewer for this comment. According to the reviewer's feedback, we investigated whether the PNIPAM solidification by salt ions forms crystalline structures or not via XRD analysis. We confirmed that PNIPAM exhibits an intrinsically amorphous polymer and does not experience a crystallization process during the solidification (Supplementary Fig. S1).

Supplementary Fig. S1. XRD pattern of (i) NIPAM, (ii) PNIPAM, (iii) NaCl, and (iv) solidified PNIPAM by NaCl, respectively. Once NIPAM dry solid powder (as received from the manufacturer) dissolved and polymerized, the PNIPAM was in an amorphous state. The PNIPAM solidified by NaCl was also amorphous, whose XRD pattern was similar to that of the PNIPAM. The sharp peaks observed in the solidified PNIPAM sample were from NaCl crystals.

(Revised manuscript, page 5) Such solidification by salt ions did not induce PNIPAM crystallization (Supplementary Fig. S1).

(Revised manuscript, Experimental section, page 17) X-ray diffraction (XRD) analysis was performed under ambient conditions in open air, using Anton Paar XRDynamic 500.

#5. The incorporation of inorganic additives such as MXene or CNT is expected to improve the thermal/mechanical stability of the composites. Therefore, I recommend conducting mechanical testing and SEM measurements of these composites to verify their physical properties and the dispersion of the nanomaterials in the polymer matrix.

Response: We thank the reviewer for this valuable suggestion. We examined the effect of MXene and CNT on the stiffness and conductivity of the solidified PNIPAM structure (Fig. 3l and 3m, and Supplementary Fig. S6). Upon contact of the PNIPAM composite solution with salt ions, composite aggregates of stiff MXene and CNT particles along with the PNIPAM chains formed (Fig. 3k), resulting in higher storage modulus compared to that resulted from the pure PNIPAM solution (Fig. 3l). Further, the solidified structure containing MXene and CNT particles exhibited a higher conductivity than that of the solidified structure of pure PNIPAM (Fig. 3m and Supplementary Fig. S6) due to the conductive nature of CNT and MXene particles.

Revised Fig. 3. **l** Storage modulus increase of PNIPAM and PNIPAM composite solutions upon the solidification by 3 M CaCl₂. **m** Conductivity of pure PNIPAM and PNIPAM/MXene+CNT composite solidified by 3 M CaCl₂ (n=4). Error bars correspond to standard deviations.

Supplementary Fig. S6. Measurement setup for the conductivity of solidified PNIPAM and PNIPAM/MXene+CNT. To demonstrate the effect of conductive inorganic particles (MXene and CNT) on the conductivity enhancement, we compared pure PNIPAM and PNIPAM/MXene+CNT composite representatively. Each solution was extruded into the 3 M CaCl₂ solution, resulting in fully crosslinked and solidified samples. Water on the surface of

the solidified samples was gently wiped using Kimtech Wipes, and the sample was placed on the customized measuring device as shown in the schematic illustration.

(Revised manuscript, page 9) The solidified PNIPAM structures were free-standable overall and could have enhanced mechanical properties and/or conductivity depending on the additive component. For example, the solidified PNIPAM/MXene (Fig. 3j) demonstrated that small particles were instantly entrapped among the aggregated PNIPAM chains in the globule state before diffusing out to the surroundings (Fig. 3k). This instant aggregation was shown as an immediate increase in storage modulus upon contact of the solution with salt ions, and the composite aggregates with stiff particles resulted in higher storage modulus (Fig. 3l). Further, the solidified structure containing MXene and CNT particles demonstrated a higher conductivity than that of the solidified structure of pure PNIPAM (Fig. 3m and Supplementary Fig. S6) due to the conductive nature of CNT and MXene particles⁴⁷⁻⁵¹.

(Revised manuscript, Experimental section, page 15) For the PNIPAM/MXene and PNIPAM/CNT solutions, the final concentration of MXene and CNT was set as the particle/polymer wt/wt ratio was 10 %, respectively. For the PNIPAM/MXene+CNT solution, the final concentration of the sum of MXene and CNT (1:1 weight ratio) was 20 %.

(Revised manuscript, Experimental section, page 17) Conductivity was measured using a 2-probe Ohmmeter mode of Keithley 2450 multimeter.

(Added references)

- 47 Shin, M. K. et al. Synergistic Toughening of Composite Fibres by Self-Alignment of Reduced Graphene Oxide and Carbon Nanotubes. *Nat. Commun.* 3, 650, doi: 10.1038/ncomms1661 (2012).
- 48 Jia, X. et al. Dramatic Enhancements in Toughness of Polyimide Nanocomposite via Long-CNT-Induced Long-Range Creep. *J. Mater. Chem.* 22, 7050–7056, doi: 10.1039/C2JM15359A (2012).
- 49 Eom, W. et al. Large-Scale Wet-Spinning of Highly Electroconductive MXene Fibers. *Nat. Commun.* 11, 2825, doi: 10.1038/s41467-020-16671-1 (2020).
- 50 Wan, S. et al. High-Strength Scalable MXene Films through Bridging-Induced Densification. *Science* 374, 96–99, doi: 10.1126/science.abg2026 (2021).
- 51 Ghaffarkhah, A. et al. Ultra-Flyweight Cryogels of MXene/Graphene Oxide for Electromagnetic Interference Shielding. *Adv. Funct. Mater.* 33, 2304748, doi: 10.1002/adfm.202304748 (2023).

We also examined whether hydrophobic CNT and hydrophilic MXene particles could be well distributed throughout the aqueous PNIPAM solution. Cross-sectional SEM images taken from lyophilized PNIPAM/CNT and PNIPAM/MXene solution displayed that hydrophobic CNT with long lengths ranging from a few micrometers formed aggregates in an aqueous PNIPAM solution and hydrophilic MXene, presented in a nanometer-sized sheet form, was well dispersed in the PNIPAM solution with minimal aggregation (Supplementary Fig. S17). Because deep studies of the effect of MXene and CNT contents and uniformity in PNIPAM solutions on the enhancement of electrical conductivity seem to be beyond the scope of this manuscript, we briefly discuss this in the manuscript. However, we are expecting that we could conduct additional studies regarding the electrical conductivity of the printed circuit/array and the development of advanced electrical devices utilizing the proposed PNIPAM-based printing techniques in the near future.

Supplementary Fig. S17. Cross-sectional SEM images of lyophilized PNIPAM/CNT and PNIPAM/MXene solutions. (a) Hydrophobic CNTs with long lengths (5–9 μm according to the manufacturer) were aggregated as denoted as arrows. (b) Hydrophilic MXene was homogeneously dispersed into the solution. As observed in these cross-sectional SEM images (in lyophilized samples), the internal structure of the PNIPAM/MXene was similar to that of the pure PNIPAM displayed in Figure 1g.

(Revised manuscript, page 14) In terms of electrically conductive structures, we could further conduct studies on the effect of the content of conductive particles (e.g., CNT and MXene particles) and on the way of obtaining homogeneous conductive PNIPAM solutions (Supplementary Fig. S17) to increase electrical conductivity for advanced electrical devices (e.g., bioelectronics).

#6. I would also recommend TGA testing of all PNIPAM-based inks to confirm their composition.

Response: We thank the reviewer for this feedback. Since the PNIPAM-based inks contain a large amount of water (80–90 wt% in total ink weight), the TGA data could not corroborate the exact ink composition information. The exact amount of each component in mol/mol % or wt/wt % for the preparation of inks was stated in the experimental section, as follows: The PNIPAM concentration was controlled from 0.5 to 1.6 M. For 1.0 M PNIPAM-based composite solutions, a MXene, CNT, PVA, PAM, or Alg dispersion/solution was mixed with the NIPAM, APS, TMEDA mixture to synthesize a PNIPAM/Dye, PNIPAM/MXene, PNIPAM/CNT, PNIPAM/PVA, PNIPAM/PAM, or PNIPAM/Alg composite solution. For the PNIPAM/MXene and PNIPAM/CNT solutions, the final concentration of MXene and

CNT was set as the particle/polymer wt/wt ratio was 10 %; the particles were not sedimented in the PNIPAM solution. For the PNIPAM/PVA and PNIPAM/Alg solutions, the final concentration of PVA and Alg was prepared as 2.3 % (wt/wt). For the PNIPAM/PAM solution, a PAM solution that was first synthesized by polymerizing AM monomer, APS (5 mol/mol% of AM), and TMEDA (4 mol/mol% of AM) was mixed with the NIPAM, APS, TMEDA mixture to obtain the PNIPAM/PAM solution comprising 1 M PAM.

#7. In the recyclability study, the authors conducted only one "print-recycle-print" cycle test. I would recommend multiple cycle tests to reveal the true potential of this technology. This is because of the possibility for fracture and fatigue of the PNIPAM chains during recycling.

Response: Thank you for the reviewer's suggestion. Accordingly, we demonstrated the recyclability of PNIPAM solution ink through five cycles in addition to the existing test result in the original manuscript (Fig. 5d). The solidified PNIPAM was able to be re-dissolved in water and then loaded into a new syringe for the next cycle of extrusion and solidification. The PNIPAM solution ink well-demonstrated recyclability through repeated PNIPAM solidification–dissolution cycles as shown in the following Supplementary Fig. S12.

Supplementary Fig. S12. Recyclability of PNIPAM ink in multiple solidification–dissolution cycles. The solidified PNIPAM in CaCl_2 was quickly rinsed with EtOH and then water to remove excessive salt ions and subsequently re-dissolved in water at a 5 °C refrigerator. The dissolved PNIPAM was loaded into a syringe and extruded into CaCl_2 for the next solidification. Such a series of procedures was repeatable. In the 5th cycle

demonstration, we added red dye for better visualization and then printed the shape of a recycle sign.

(Revised manuscript, page 11) PNIPAM recycling was also possible by simply rinsing and dissolution in water without the drying process (Supplementary Fig. S12). The solidified PNIPAM fully dissolved in water at low temperatures (below the LCST of PNIPAM solidified by salt ions) and then was extruded into a salt solution for solidification. This solidification–dissolution cycle was repeatable.

REVIEWERS' COMMENTS

Reviewer #1 (Remarks to the Author):

I thank the authors for addressing all of my concerns. The manuscript is much better than the original draft and I recommend publication.

Reviewer #2 (Remarks to the Author):

The authors have addressed all the issues I raised and have revised the manuscript accordingly. Therefore, I recommend acceptance of this manuscript for publication.

Responses to Reviewers' Comments

The following is the Response to the Reviewers' Comments for the manuscript entitled "Sustainable 3D printing by reversible salting-out effects with aqueous salt solutions" (NCOMMS-23-44607) to *Nature Communications*.

Reviewer #1 (Remarks to the Author)

: I thank the authors for addressing all of my concerns. The manuscript is much better than the original draft and I recommend publication.

Response: We appreciate the Reviewer's valuable suggestions. We could have improved the manuscript quality significantly.

Reviewer #2 (Remarks to the Author)

: The authors have addressed all the issues I raised and have revised the manuscript accordingly. Therefore, I recommend acceptance of this manuscript for publication.

Response: We appreciate the Reviewer's valuable suggestions, as they have helped us to improve the manuscript quality significantly.